# Patterns and comparisons of human-induced changes in river flood impacts in cities

Stephanie Clark[1*], Ashish Sharma[2], Scott A. Sisson[1]

**1:** School of Mathematics and Statistics, University of New South Wales, Sydney, Australia
**2:** School of Civil and Environmental Engineering, University of New South Wales, Sydney, Australia
***** *corresponding author*

## ABSTRACT

In this study, information extracted from the first global urban fluvial flood risk data set (Aqueduct) is investigated and visualised to explore current and projected city-level flood impacts driven by urbanisation and climate change. We use a novel adaption of the self-organizing map (SOM) method, an artificial neural network proficient at clustering, pattern extraction and visualisation of large, multi-dimensional data sets. Prevalent patterns of current relationships and anticipated changes over time in the nonlinearly-related environmental and social variables are presented, relating urban river flood impacts to socioeconomic development and changing hydrologic conditions. Comparisons are provided between 98 individual cities. Output visualisations compare baseline and changing trends of city-specific exposures of population and property to river flooding, revealing relationships between the cities based on their relative map placements. Cities experiencing high (or low) baseline flood impacts on population and/or property that are expected to improve (or worsen), as a result of anticipated climate change and development, are identified and compared. This paper condenses and conveys large amounts of information through visual communication to accelerate the understanding of relationships between local urban conditions and global processes.

**Keywords:** urban hydrology; self-organizing maps; SPATIOTEMPORAL patterns; clustering; city-scale; neural networks; Anthropocene

# 1 INTRODUCTION

The hydrologic regimes producing urban floods within many cities are varying due to anthropologic influences such as climate change and urban development (Revi et al., 2014; Mills, 2007; Desai et al., 2015; UNEP, 2016; Willems et al., 2012; Desai et al., 2015), as the high densities of population, property, infrastructure and industry within cities are both substantial drivers and receivers of environmental impacts. River flooding currently impacts more people than any other environmental event (Doocy et al., 2013; Desai et al., 2015; Sofia et al., 2016) posing a threat to almost 380 million urban residents globally (UN-Habitat, 2014). However, most existing river flood assessments are at a local or regional scale (as in Muis et al., 2015), limiting the possibility to compare between multiple cities, and studies at a global scale have traditionally been limited by a lack of datasets and methods (Jongman et al., 2012). The first unified global set of urban fluvial flood risk data has recently been published for a selection of cities as part of the World Resources Institute's Aqueduct Global Flood Analyzer Tool (subsequently referred to as Aqueduct) (Winsemius et al., 2013; Ward et al., 2013; website 3), consisting of current and projected future flood impacts on population and property resulting from climate change and urbanisation. Whilst this data set has previously been analysed and visualised by its creators at the watershed scale (Winsemius et al., 2016), the large amount of city-level information is currently presented on an individual city basis. Here we conduct an analysis, including visualisation and clustering using the self-organizing map (SOM) technique, of the total collection of data that is available at the city scale, to investigate the inherent relationships and patterns amongst the city data. A type of artificial neural network, the SOM is useful for exploring relationships between nonlinearly related variables, and is popular for investigating potentially difficult-to-define environmental responses to human influences (e.g. Shanmuganathan et al., 2006; Vaclavik et al., 2013; Clark et al., 2016b) as well as providing comparisons between geographic areas (Kaski & Kohonen, 1996; Clark et al., 2015; Clark et al., 2016). We expand the traditional SOM method to allow for a comparison over time of expected changes in the city-level conditions. The aim of this study is to increase the understanding of prevalent global patterns of human-environmental relationships influencing city-level river flooding, and discover how a global set of individual cities fits into these patterns of shifting quantities of water and city features.

The combined effects of climate change and urbanisation (alterations in urban population growth, development, land use and density) are leading to variations in the magnitude, frequency and timing of precipitation, snow melt and river floods within cities, generally producing higher peak river flows with shorter response times (Frich, et al. 2002; Desai et al., 2015; UNEP, 2016; Wasko & Sharma, 2015; Shiermeier, 2011; Cunderlik, 2009; Barnett et al., 2005; Immerzeel et al., 2010). The changing patterns of precipitation and runoff are complex and not uniformly spatially distributed (Meehl et al., 2005; Desai et al., 2015; Wentz et al., 2007; Frich et al., 2002). Increases in rainfall intensity at urban hydrology scales of up to 60% are anticipated by 2100 (Willems et al., 2012), and the micro-climates of cities are expected to interact with climate change in a number of ways that will potentially exacerbate flood effects (Revi et al., 2014). Highly populated urban areas are experiencing an increase in flood vulnerability through migration into urban flood plains (Kreimer et al., 2003; Jongman et al., 2012; Revi et al., 2014), causing the global population exposed to river flooding to increase faster than total global population growth (Jongman et al., 2012). By 2050, it is estimated that 70% of the world's population will live in cities (UN-Habitat, 2010), up from 54% in 2015 (UN-DESA, 2015), and as cities grow the proximity of population and property to water courses will continue to increase (Kummu et al., 2011). Urban land cover is increasing globally, with more impervious areas and higher building densities, at a rate over double that of urban population growth (Angel et al., 2010a), and is projected to increase three-fold by 2030 (Pachauri et al., 2014). In the future, cities in particular are predicted to become even more vulnerable to extreme hydrologic events (Pachauri et al., 2014; Willems et al., 2012; Revi et al., 2014, Sofia et al., 2016), and an understanding of each city's unique response to these

hydrologic and developmental changes will be necessary as cities strive to adapt (Revi et al. 2014; Doocy et al.,

2     2013).

In this paper, a visual comparison is produced amongst a selection of cities based on their current and projected
future urban river flood impacts on population and property, resulting from an anticipated combination of
climate change and development. It should be noted that fluvial flooding is the only type of flooding that is
considered here, and this study does not include an analysis of cities subject to coastal or pluvial flooding.
Analysing data with city-specific projections of changes in hydrology, population and development levels (based
on future climate scenarios, projected development pathways, and a best assumption of flood protection
standards) we produce an analysis and visualisation of the patterns of baseline conditions and anticipated
changes in city-level river flooding impacts to the year 2030. We establish the prevalent global spatial and
temporal patterns of urban flood impacts, explore these impacts as resulting from both developmental and
hydrological drivers, and match the cities to their most similar pattern. The patterns are established through
dimension reduction, clustering and visualisation of multivariate data with an adaptation of the self-organizing
map technique. We begin by presenting analyses of patterns of urban flood conditions (as measured by the
amount of population affected and urban damages costs) for a baseline global snapshot (2010), then investigate
projected temporal changes (up to 2030), and finally combine this information into a global temporal analysis of
the cities. As individual cities are matched to their closest patterns at each stage, we discover clusters of cities
with similar urban flooding characteristics and projected trends.
The main analyses to date on this important data set (Winsemius et al., 2013; Ward et al, 2013) have been
conducted by its creators, in which the data has been analysed at the watershed scale. Though this data set has
been published at city scale as well as watershed scale, certain limitations do exist to an analysis at city scale. The
spatial scale of the applied hydrologic and hydraulic models used to create the data, the potentially small
upstream catchment areas of the cities, the unidentified relationships of cities along the same river in which
mitigation efforts upstream may effect downstream cities, and the assumption of city-specific defence measures
(which are discussed in the 'Data' section) all increase the uncertainty inherent in these hydrologic estimates
more so than if the data were analysed at the basin scale. However, an analysis of the city data is important in
allowing an understanding of conditions that can be expected on a city-level and the identification of cities
potentially facing similar circumstances, since climate change, development, and urban administrations are not
restricted to river basin boundaries. Therefore, we accept these inherent limitations in order to discover what
insights the most cohesive city-level data set to date may reveal.
A growing body of research is investigating the impact of anthropogenic changes on urban flooding at regional
and global scales, however we have found no literature comparing specific cities in terms of changing city-level
flood impacts on populations and property. The Intergovernmental Panel on Climate Change's 5[th] Assessment
Report Chapter 8 'Urban Areas' (Revi et al., 2014) discusses the vulnerabilities and resilience of cities to climate
change in general, noting that the analysis is based on economic losses and would differ if a human component
is included. Jongman et al. (2012) investigated global trends of coastal and river flooding based on changing
regional population densities and land use. Increased vulnerability to flooding is attributed to population growth
or increases in wealth, though the modelling does not include changing hydrology due to climate change.
Jongman et al. (2015) estimated regional trends in human and economic river flooding vulnerabilities by income
level, through hazard and exposure calculations. Kunkel et al., (1999) investigated the increasing trend of
economic losses and fatalities in the USA due to increasing vulnerability to floods, however the climate change
contribution to this increase was not possible to quantify due to a lack of data. Winsemius et al. (2016) produced
the first projections of global future flood risk that consider separate impacts of climate change and
socioeconomic development, with results discussed by geographic region (river basin) and economic level. The
investigation of the connection between coastal flooding and climate change (increasing storms combined with
sea level rise) is more common in the literature than the connection between river flooding and climate change
(Nicholls et al., 2008; Nature, 2016) due to better data availability. Sofia et al (2016) emphasize that analyses of
climate change and socio-economic development as both drivers and receptors of flood risk is needed. Muis et
al. (2015) call for an investigation between the combination of land use change and hydrologic change on future
flood risk. Jongman et al. (2012) highlight that due to population growth and climate change, global methods
incorporating both spatial and temporal dynamics to investigate inland flooding at the city scale are necessary
for global development studies and estimating costs associated with climate change. To date, a global
examination of changing flood conditions at the city level resulting from urban development and climate change,
including a direct comparison between specific cities, has not been made. The analysis we present here
corresponds directly to this gap in the literature.
General patterns as well as specific relationships can be extracted from the output maps in this paper. In the
interest of channelling the 'potential of visual communication to accelerate social learning and motivate
implementation of changes' (Sheppard, 2005) the aim of the method used here is to discover and demonstrate
potentially interesting global patterns and relationships that would not otherwise be evident in the data, for
example: clusters of cities which are currently experiencing high flood impacts that are projected to greatly
increase in the future, and to what extent this may be due to climate change (or socioeconomic development)
within each city; which cities not currently experiencing notable effects of flooding may expect to in the future;
which cities are projected to mitigate potentially adverse flood effects from climate change with reductions in
flooding due to socioeconomic factors; which cities are projected to experience an increased flood vulnerability
driven by socioeconomic factors alone; and the relationship between the changes in vulnerability of the
population and urban damages costs for each city.
The comparison of individual cities in this study (rather than river catchments) allows a blending of
environmental and social information which reinforces the co-dependence of humans and their natural
environment, a relationship which is often easily overlooked by urban dwellers. Explicitly visualising the role that
urbanisation may have on the environmental conditions experienced by urban citizens is an essential reminder
of this connection. Cities potentially facing similar circumstances and challenges are identified in this study,
suggesting possibilities for a sharing of strategies. As climate change, development, and urban administrations
transcend river basin boundaries, an investigation of impacts and determination of potential mitigation
strategies at the city level as well as the basin level expands the potential for decision makers to be presented
with all the available, relevant data for consideration.

## 2 DATA AND METHOD

### DATA

The data set used in this study combines city-level estimates of annual expected urban river flood impacts on
population and urban damages costs (2010), projections of future changes in flood impacts attributed to climate
change and/or development (up to 2030), and socioeconomic data for a globally distributed set of cities.
The selection of cities used here is based on a list provided by the Lincoln Institute of Land Policy's Atlas of Urban
Expansion (Angel et al., 2010, website 1), spanning all continents except Antarctica, encompassing four economic
levels and four population levels. City population data (2010) and future population estimates (2030) are from
the UN Department of Economic and Social Affairs (UN-DESA, 2015), and GDP per country are from the World
Bank's World Development Indicators database (website 2).
Annual river flood impact estimates are obtained from Aqueduct, the global dataset of fluvial flood risk. As this
data is solely related to the influence of fluvial flooding on metropolitan areas, it does not include coastal or
pluvial flood risks. In this data set, Aqueduct provides separate estimates of annual impacts on the number of
affected population (people exposed to flood waters) and urban property damages costs (in US dollars), which
will be referred to in this paper as 'population' and 'damages' impacts.
Global hydrologic and hydraulic models, inundation modelling, and spatial data sets of population, land use and
infrastructure are used within Aqueduct to quantify flood risk in each city. Aqueduct identifies future anticipated
changes in urban flood vulnerabilities as driven by climate change (altered hydrology), socioeconomic
development (population, land use and economic changes), or in most cases a combination of both. Either of
these drivers may increase or decrease the frequency and intensity of flooding, and the resulting flood impacts,
for a given city. Three separate scenarios of climate change and socioeconomic development (optimistic,
business-as-usual, and pessimistic) are given in Aqueduct, and in this study we use data from the business-as-
usual case for our future flood impact scenario. Future hydrologic and hydraulic estimates in Aqueduct are based
on global circulation model data from the ISIMIP project (website 4) and changes in population and economic
development are based on Shared Socioeconomic Pathways data with a downscaling procedure that
differentiates between urban and rural growth (website 5; Samir & Lutz, 2014). Recent papers published with
this data include Winsemius et al. (2016), Jongman et al. (2015) and Muis et al. (2015).
Expected flood impacts are provided by Aqueduct for nine possible levels of city-wide flood protection, from
protection against the 2-year average return interval (ARI) flood to the 1000-year ARI flood. This protection level
indicates how well protected the area is against flood damage, based on the standard or capacity of flood
protection measures such as dikes, levees or dams. In this study, we assign an assumed flood protection level to
each city based on the country's World Bank income level (as in the World Resource Institute's Aqueduct Global
Flood Risk Country Rankings, website 6) due to a lack of information on each city's actual protection level. This
method follows recommendations based on the rational that higher standards of protection against flooding
may be expected in higher income countries (Jongman et al., 2012; Nicholls et al., 2008), and findings by Doocy
et al. (2013) that flood impacts are significantly associated with classification of income level by the World Bank.
We assume each city's flood protection level remains the same during the timeline of this study.
To allow for a comparison between cities of greatly differing sizes and hydrologic conditions, the wide-ranging
data values were log-transformed. The data set was then standardized by transforming these values linearly into
the range 0-1 (with the lowest value becoming 0 and the highest value becoming 1) for each variable (population
affected, urban damages, etc). The data is log transformed, following recommendation by Agarwal & Skupin
(2008) that highly skewed variable distributions may benefit from log transformation before use in the SOM.
Cities with no flood impacts in both 2010 and 2030 were removed (22 cities), though cities with no flood impacts
in 2010 but with flood impacts in 2030 have been kept in the study. The final list of cities is presented in Table 1.
**TABLE 1: CITY LIST -** alphabetically by region.

| Eastern Asia & the Pacific | | | | | |
|---|---|---|---|---|---|
| Anqing | China | Seoul | Rep. of Korea | Bandung | Indonesia |
| Ansan | Rep. of Korea | Shanghai | China | Bangkok | Thailand |
| Beijing | China | Sydney | Australia | Ho Chi Minh City | Vietnam |
| Changzhi | China | Tokyo | Japan | Kuala Lumpur | Malaysia |
| Chinju | Rep. of Korea | Ulan Bator | Mongolia | Manila | Philippines |
| Fukuoka | Japan | Yiyang | China | Palembang | Indonesia |
| Guangzhou | China | Yulin | China | Songkhla | Thailand |
| Leshan | China | Zhengzhou | China | | |
| Pusan | Rep. of Korea | | | South Asia | |
| | | Southeast Asia | | Dhaka | Bangladesh |

| City | Country | | City | Country | | City | Country |
|------|---------|---|------|---------|---|------|---------|
| Hyderabad | India | | Aswan | Egypt | | Sao Paulo | Brazil |
| Jalna | India | | Cairo | Egypt | | Tijuana | Mexico |
| Kanpur | India | | Casablanca | Morocco | | Valledupar | Colombia |
| Kolkata | India | | Marrakech | Morocco | | **North America** | |
| Mumbai | India | | Port Sudan | Sudan | | Chicago | United States |
| Puna | India | | Tebessa | Algeria | | Cincinnati | United States |
| Rajshahi | Bangladesh | | | | | Houston | United States |
| Vijayawada | India | | **Sub-Saharan Africa** | | | Los Angeles | United States |
| | | | Accra | Ghana | | Minneapolis | United States |
| **Western & Central Asia** | | | Bamako | Mali | | Modesto | United States |
| Ahvaz | Iran | | Harare | Zimbabwe | | Philadelphia | United States |
| Astrakhan | Russian Fed. | | Ibadan | Nigeria | | Pittsburgh | United States |
| Baku | Azerbaijan | | Johannesburg | South Africa | | Springfield | United States |
| Gorgan | Iran | | Kampala | Uganda | | St. Catharine's | Canada |
| Istanbul | Turkey | | Kigali | Rwanda | | Tacoma | United States |
| Kuwait City | Kuwait | | Ouagadougou | Burkina Faso | | | |
| Malatya | Turkey | | | | | **Europe** | |
| Moscow | Russian Fed. | | | | | Budapest | Hungary |
| Oktyabrsky | Russian Fed. | | | | | Castellon | Spain |
| Sanaa | Yemen | | **Latin America & the Caribbean** | | | Le Mans | France |
| Shimkent | Kazakhstan | | Buenos Aires | Argentina | | Leipzig | Germany |
| Teheran | Iran | | Caracas | Venezuela | | London | UK |
| Tel Aviv | Israel | | Guadalajara | Mexico | | Madrid | Spain |
| Yerevan | Armenia | | Ilheus | Brazil | | Paris | France |
| Zugdidi | Georgia | | Jequie | Brazil | | Sheffield | UK |
| | | | Mexico City | Mexico | | Thessaloniki | Greece |
| **North Africa** | | | Montevideo | Uruguay | | Warsaw | Poland |
| Alexandria | Egypt | | Ribeirao Preto | Brazil | | Wien | Austria |
| Algiers | Algeria | | Santiago | Chile | | | |

## METHOD

We use an extension to the self-organizing map method to determine patterns and similarities in the impacts, changes and drivers of urban flooding amongst the cities. The self-organizing map (SOM, Kohonen, 2001) is an unsupervised learning algorithm from the family of artificial neural networks that discovers patterns in multivariate data sets with nonlinear inter-variable relationships.

The SOM reduces the dimensionality of the data set by creating a (in this case) two-dimensional map grid which, through an iterative process, is essentially bent and stretched over the data set until it best characterizes the shape of the data cloud. The numerous data items become represented by a (usually) much smaller number of map nodes, known as prototypes. The map nodes, or prototypes, move iteratively into position amongst the data whilst maintaining their grid formation, establishing a higher density of prototypes in areas of higher data density. Once in position, the prototypes represent the most prevalent patterns in the data. Each data item is then matched to its closest prototype, creating clusters of similar data items.

The SOM algorithm consists of a two-step iterative process of comparing the map and the data, and then updating the map to better represent the data. The method begins with a calculation of distances in data space (in this case we use Euclidean distance) between each data item, $x_i$ (where $i = 1:N$), and each map node, $m_j$ (where $j = 1:M$). Data and map nodes vectors are all of the same dimension, $d$. The goal of the comparison stage is to find the nearest map node to each data item (commonly referred to as the best matching unit, BMU), which is then given the index $c$, using the following calculation:

$$\|x_i - m_c\| = \min_j \{\|x_i - m_j\|\}.$$

This partitions the data into subsets of items sharing the same nearest node, $m_c$. Next, the locations of the map nodes are adjusted to become closer to their nearby data items. Application of a smoothing

‘neighbourhood’ kernel during this stage produces a smoother map by updating neighbouring nodes to a similar extent based on the nearby data. That is, the location of each map unit, $m_j$, becomes updated based on a weighted average of the data items matching itself as well as its neighbouring nodes, where the weighting is given by the neighbourhood kernel. The size of the kernel decreases with each iteration to include fewer nodes. We use a Gaussian shaped neighbourhood kernel, where $h_{ij}$ (the neighbourhood kernel element indicating the influence of each data item, $x_i$, on the updating of node $m_j$) is defined at iteration $t$ as:

$$h_{ij}(t) = \exp(\frac{-(m_c - m_j)^2}{2\sigma^2(t)})$$

where σ is the kernel radius. At each iteration (t), the updated node locations are found as in (Kohonen, 2013):

$$m_j(t+1) = \frac{\sum_{i=1}^{N} h_{ij}(t)\, x_i}{\sum_{i=1}^{N} h_{ij}(t)}.$$

After map training is complete, the map node vectors each represent a unique combination of variables in the data, according the final location of the map nodes in data space. Each of these unique combinations of variables represent a characteristic pattern in the data. The data items are once again matched to their closest map node, forming clusters of data that best match each pattern.

In this study, the 'patterns' are the key characteristics represented by each map node vector (such as specific baseline and/or projected flood conditions, and the drivers of change). The 'cluster' members are the cities that match the pattern represented by their nearest map node better than they match the patterns of any other nodes.

As the SOM is an unsupervised learning algorithm, there is no subjectivity in the resulting cluster memberships. The iterative training process discovers the principal curves of the data set (the nonlinear directions of maximum variance) and aligns the map coordinate system with these, so that the two axes of the map generally follow the first two principal curves of the data. When the map is presented in its two-dimensional form, with data items located at their nearest map node, similar data ends up in close proximity on the map and dissimilar data is far apart. Through the SOM creation process the prevalent data patterns are identified by the nodes, data items become grouped into clusters around these patterns, and the clusters are ordered by similarity on the map. For a more detailed summary of the SOM method, refer to e.g. Clark et al. (2015).

Error measures, such as quantization error (QE), topographic error (TE) and dimension range representation measure (DRR) are used to compare the data set and maps.

The QE (Kohonen, 2001) measures how well the map nodes represent the data items using the sum of squared Euclidean distances between each data item, $x_i$, and the node closest to it, $m_c$, averaged over all data points:

$$QE = \frac{1}{N}\sum_i \|m_c - x_i\| = \frac{1}{N}\sum_i \sqrt{(m_c^2 + x_i^2 - 2m_c x_i)}.$$

The TE (Kiviluoto, 1996) indicates how well the topography of the data set is preserved on the map, giving higher error values for maps that are unnecessarily bent or twisted. The BMU and second BMU for each data point are checked to determine if they are adjacent ($u_{x_i} = 1$ if the first and second BMUs of $x_i$ are neighbours, 0 otherwise), and TE is calculated as:

$$TE = \frac{1}{N}\sum_{i=1}^{N} u_{x_i}$$

The DRR (Clark et al., 2015) measures how well the map represents each variable of the data set to ensure even
coverage of the dimensions. The maximum intra-cluster spread of data items in each dimension, $d$, that become
represented by a single map node, $x_i$ (as a proportion of the overall data range in that dimension) is determined.
The DRR is calculated as follows, where $x_i(d)$ are data values in dimension $d$, and $x_{ij}(d)$ are the data values in
dimension $d$ that are assigned to map unit $j$:

$$DRR(d) = \max_{j} \frac{\max_{ij}(x_{ij}(d)) - \min_{ij}(x_{ij}(d))}{\max_{i}(x_i(d)) - \min_{i}(x_i(d))}$$

In this study, the data set is split into two subsets ('baseline' data and 'projected future changes') for each city,
allowing a progressive investigation of spatial and temporal patterns of urban flooding. A series of three
separate SOMs (also referred to as maps) are created with prevalent global patterns and city similarities
established separately on each map through colouring and labels, as follows:

- SOM1 explores the spatial properties of the baseline data set, enabling a comparison of the state of urban river flood impacts in each city at a snapshot in time (2010).
- SOM2 explores patterns of projected temporal changes in impacts of urban flooding on population and property (to 2030), incorporating the drivers of climate change and urban development, and
- SOM3 portrays the temporal relationships between the cities in a type of longitudinal exploratory data analysis, clustering cities that are similar in the baseline situation and are also projected to trend similarly in response to each driver in the future.

SOM1, the baseline map, depicts prevalent global spatial patterns and identifies urban flooding conditions in
each city based on two variables: 1) the total population affected annually by river flooding, and 2) annual urban
property damages costs incurred by river flooding. The map is created based on these two variables, though by
projecting new variables onto the trained map it is also used to show: 3) the percentage of each city's population
affected, and 4) the percentage of the country's GDP affected. Usually used with higher-dimensional input data,
the SOM method is useful here for creating a map with two variables as the nonlinear projection establishes the
relationships between cities in alignment with the directions of maximum variance (ie. the directions of most
importance) in the data. It also allows for the results to be used as input into SOM3 later.
SOM2, the future projected changes map, describes the anticipated alterations in urban river flooding in each
city by 2030. This map is based on four variables of projected changes and their associated drivers: 1) the
projected change in population affected annually, 2) the projected change in annual urban damages costs, 3) the
proportion of change in population affected that is anticipated to be attributable to climate change, and 4) the
proportion of change in urban damages costs that is anticipated to be attributable to climate change. The
remainder of the increase or decrease in impacts is attributed to socioeconomic causes (such as population
change, urban density change, increased city footprint, and changes in urban land cover).
SOM3, the temporal map, uses the location of each city along the axes of the two-dimensional baseline and
future projected changes maps (which essentially delineate the first two principle curves in each higher
dimensional data subset) as input data. In creating SOM1 and SOM2, the baseline and future data subsets have
already been reduced to their two most prominent dimensions respectively (which have become the axes of
these maps), and each of these four dimensions is considered equally when placing the cities on the temporal

map. This method is based on the method used in Clark et al. (2015) to investigate individual data items transitioning through a self-organizing-time-map, and has been modified for the comparison of patterns on two-dimensional maps of differing sizes and shapes that have been created separately based on different variables.

Distinct patterns that have emerged through the process of training the three maps are represented by the nodes of SOM3. These patterns are the most relevant combinations of dynamic city flood impacts, socioeconomic, and climate change characteristics in the overall data set. SOM3 is clustered, coloured and labelled to indicate the relationships between the cities in terms of similar or differing baseline situations *and* projected changes. Cities with relatively close locations on both the baseline and future projected changes maps are considered to have parallel temporal paths, and will be found close together on the temporal map. Those with converging trends (dissimilar baseline conditions, but similar future projected changes) and diverging trends (close baseline conditions, but dissimilar future projected changes) are also identifiable on this map.

In the creation of each map, grid size and shape have been determined using QE, TE, and DRR, with comparisons between the data set and each potential map. For the baseline map, a 10*7 grid is found to be the optimum shape to represent the data based on the error measures. An 8*8 map is fitted to the future projected changes data set. After finding these optimum side ratios, the maps are increased in size preserving their side ratios (to 20*14 and 18*18) to allow the data items to spread out until most cities are placed individually, allowing the relationships between all cities to become evident (as in Skupin & Hagelman, 2005). The temporal map is sized at 25*17 nodes. Whilst the input data for the baseline and future projected changes maps were standardized into the range 0-1 before training, the input data for the temporal map is not standardised in order to preserve the ratios between the lengths of the first two principal curves in each of the first two data subsets.

Prevalent cluster characteristics are determined using a 'second level' clustering of the nodes of the SOM (as in Vesanto & Alhoniemi, 2000; Skupin & Hagelman, 2005), performed using Ward's clustering method (Ward, 1963) with the number of clusters determined using the Davies-Bouldin index (Davies & Bouldin, 1979). The Davies-Bouldin index reports the ratio of within cluster scatter ($S_j$ for cluster $j$) to inter-cluster distances, looking at each cluster and its most similar one, ($M_{jk}$), with a lower ratio ($S/M$) indicating a better estimate of the number of clusters of interest present in the data. Ward's minimum variance method is a hierarchical clustering algorithm based on minimizing the total within-cluster variance. With this second-level clustering, each data item of the original data set becomes a member of the same final cluster as its closest node (Vesanto & Alhoniemi, 2000).

The final clustering is visually verified with a SOM 'U-matrix' (Ultsch, 2003). The U-matrix visualises distances in data space between immediately neighbouring nodes, indicating these distances by colour on a grid of the same size as the SOM. By computing how close adjacent map nodes are in data space, the U-matrix is able to provide an indication of cluster boundaries based on large dissimilarities between neighbouring nodes. A greater change in relative distance between the locations of the nodes in data space than in map space is displayed in a lighter colour on the grid, and lesser distances in darker shades. The darker regions of the grid then indicate the cluster centres, separated by lighter coloured boundary areas.

By reducing the information from this multivariate data set into the two most prominent dimensions and finding relationships between the data items at each of these three stages, spatial and temporal information about global patterns of urban flooding is abstracted, and similarities and differences between the cities are clearly portrayed. This method extracts two levels of information:

(1) the most characteristic socio-environmental patterns in the data are found, and
(2) cities are compared to each other with respect to their relative flooding conditions.

The simulations are run in MATLAB with use of the SOM Toolbox (website 7) with variables and map sizes as described above.

## 3 RESULTS

Three SOMs are presented sequentially to reveal three unique sets of patterns in the data, where the term 'patterns' refers to combinations of variables that characterise a specific set of conditions. The cities are clustered into groups with conditions matching these patterns, based only on the given data. The maps each have different sizes, shapes and colours as they represent different subsets of input data. The interpretation of the maps is discussed in this section.

### SOM1: BASELINE URBAN FLOOD IMPACTS

Patterns of urban flood conditions in 2010 are shown on the baseline map, SOM1, in Figure 1. The placement of city labels indicates the relationship of each city to each other in terms of river flood impacts on population and urban damages costs. The map is created by organizing the cities with respect to each other based on both of these factors. Cities close together are more similar in the amount of population affected and urban damages costs, and cities located far apart are less similar.

The relative placement of the cities on the map is the main map characteristic providing insight into the features of the data, indicating differences in a *combination* of the variables which can be discerned from the colouring of Figure 1(a). Each map node has a four-component vector (representing the value of each of the four variables at the location of the node in data space). The four images in Figure 1(a) show SOM1's city labels over grids coloured separately by the values of each of the four variables (white is low, purple is high). For each city, the relative value of each of the variables can be seen. For example, Cincinnati (top right) incurs high material damages costs, and medium population affected, whereas Ulan Bator (mid left) has similar population affected to Cincinnati, but much lower material damages costs.

The nonlinearity of the relationships between the variables is evident from the colouring of the grids, indicating that high (or low) values are located on different regions of the map for each variable. The smooth transition of the values of each variable is also apparent by the smooth transition of colours along the grids. General information about the prevalent baseline global patterns and the relative flood conditions in the specific cities can be gained from inspection of these map labels and coloured grids.

Each area of the grid represents a general pattern, or combination of variables in the data, some of which are indicated by annotations on Figure 1(b). In general, higher amounts of population affected and urban damages costs resulting from river flooding are represented by areas towards the top of the map, and these variables decrease in value down the map. Values of affected population are lowest just in from the lower left corner and undulate along the bottom of the map, sweeping upwards to a maximum at the upper left corner. Urban damage values are lowest in the lower left corner and increase in concentric arcs up to the upper right corner. Generally, the left of the map contains patterns involving higher impacts on populations than on property, and the right of the map higher impacts on property than on populations.

1    a)

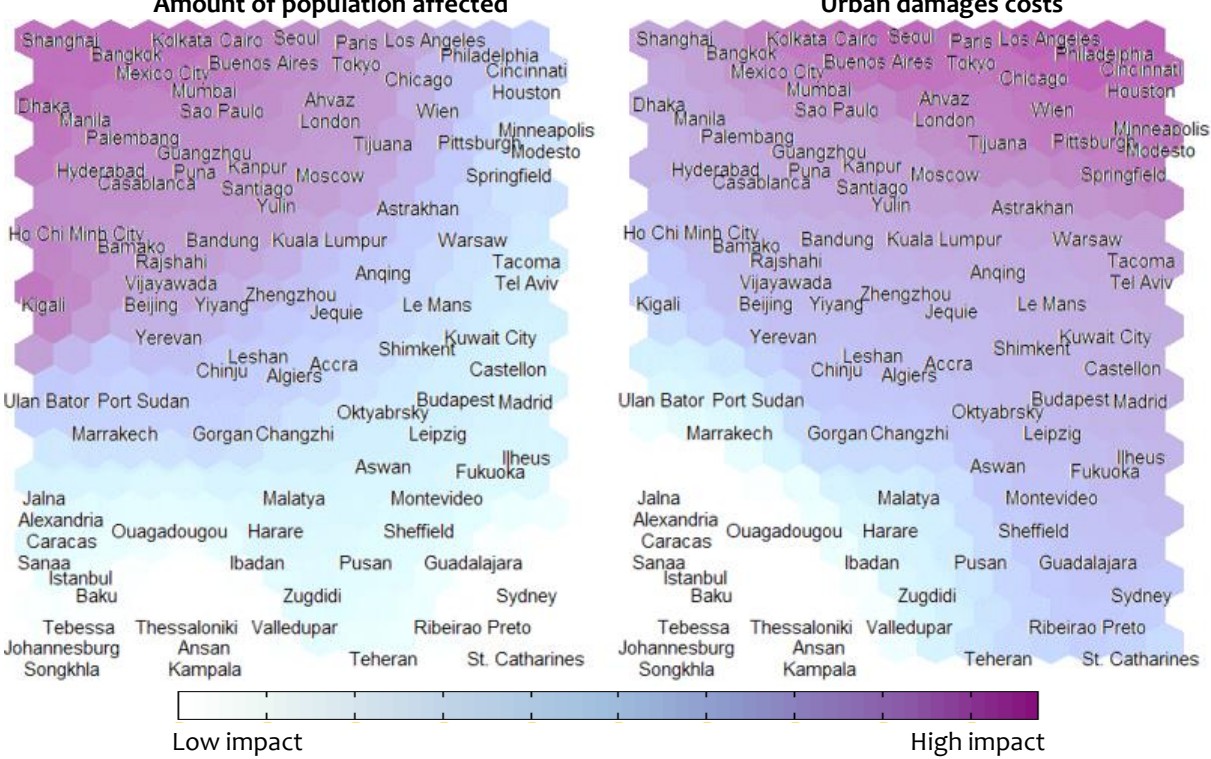

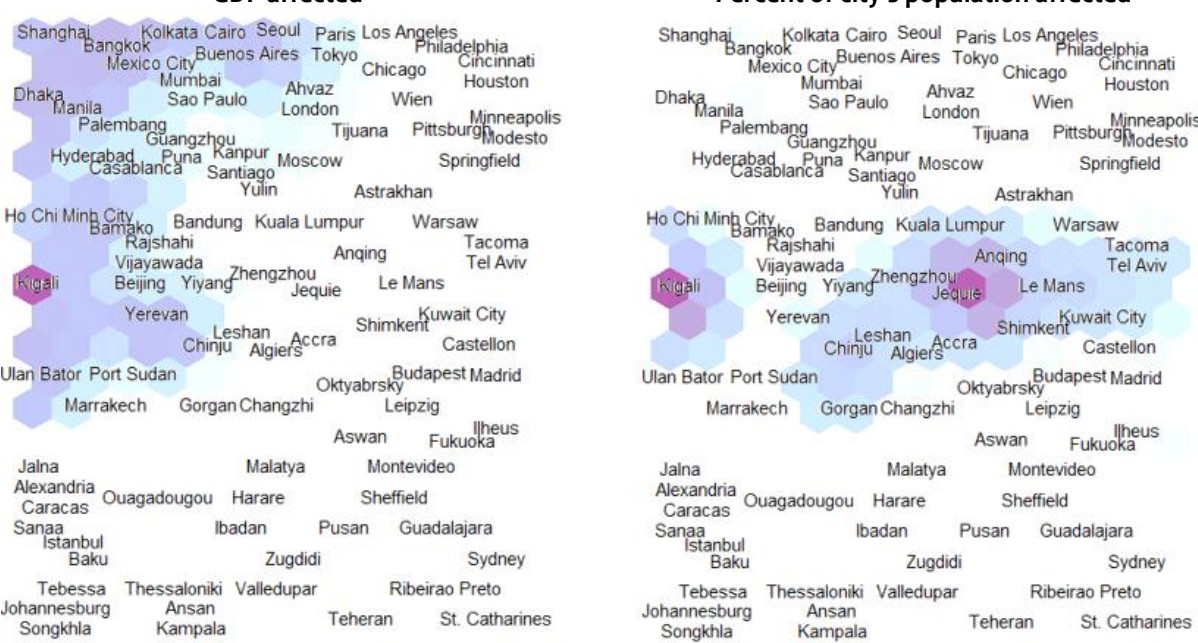

b)

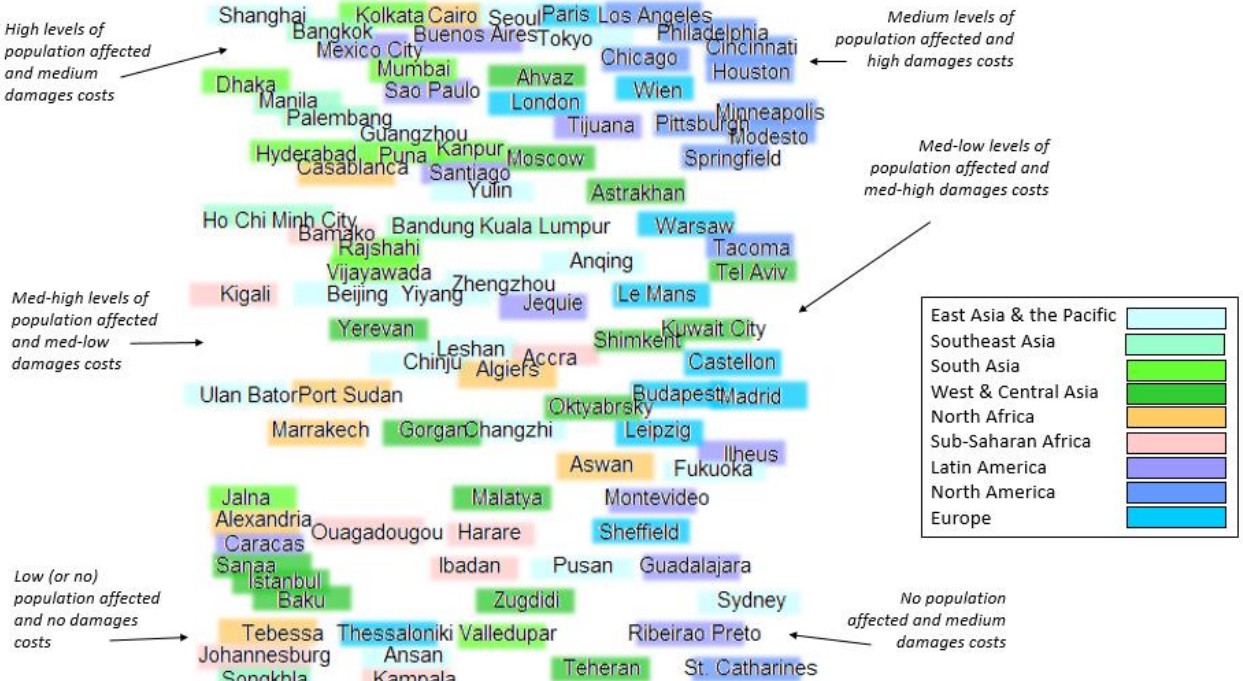

FIGURE 1: SOM1 - BASELINE (2010) URBAN FLOOD CONDITIONS. Cities are placed relative to each
other based on annual river flooding impacts on population and urban damages costs. a) The same
map is repeated for each of four variables, with colouring indicating low (white) and high (purple)
values. b) The city labels are coloured by region (see Table 1), and characteristic patterns of
general areas of the map are annotated. The reader may refer to the online version to zoom
in on text if required.

From Figure 1(b), relationships can be discerned between regions, as well as between cities in the same region.
For instance, cities in North Africa, Sub-Saharan Africa and West & Central Asia are predominantly located in the
lower portion of the map, corresponding to a prevalent pattern of low flood impacts on both population and
property. Cities in Southeast and South Asia generally correspond to the patterns of high impacts on population
and property found in the upper left of the map. Cities in Europe stretch from the top to the bottom of the map,
ranging from high overall flood effects (Paris) to no flood effects at all (Thessaloniki). North American cities are
matched to patterns that represent more significant impacts on property than on population (down the right
side of the map), and are split between those with high property damages (Philadelphia, LA, etc. – in the top
right) and those with low damages (St. Catherine's – in the bottom right).

Impacts on GDP and the proportion of the cities' populations affected are shown in the two lower maps of Figure
1(a), though these variables were not used to position the cities on the map. Cities in which river-related urban
flooding is estimated to highly affect the country's GDP are coloured on the lower left map. Kigali, in particular,
which incurs medium-high flood impacts, sees a large impact on Rwanda's GDP, perhaps because Kigali is the
main city in this relatively small country (Kreimer et al., 2003).  GDP is most affected by flooding in: Kigali,
Bangkok, Yerevan, Dhaka, Bamako and Cairo. Cities in which the flood-affected population forms a significant
proportion of the city's population are coloured on the lower right map, predominantly in a horizontal strip

across the centre. The highest proportions are in: Jequie (15%), Kigali (7%), Chinju (6%), Le Mans (5%) and Tacoma (3%).

## SOM2: Projected changes in urban flood impacts (to 2030)

SOM2 identifies the projected patterns of evolving river flood conditions in the cities (between 2010 and 2030), based on city-specific projections of increasing or decreasing flood impacts on population and damages costs, and whether these changes are anticipated to be driven more by climate change or development (Figure 2).

In Figure 2(a), regions of the map representing projected increases in flood impacts on either populations or damages costs are coloured blue and reductions in flood impacts are coloured brown (in the top row), with white indicating no projected change. Projected changes primarily driven by socioeconomic development are coloured purple (in the lower row), and green indicates that the primary driver is climate change. White represents a mid-point in which both climate change and development are predicted impact future flood conditions relatively equally. Areas of the map representing patterns of increased flood impacts predominantly due to climate change or development can be located on Figure 2(b).

Investigating SOM2, we see that climate change is projected to be predominantly responsible for increases in population vulnerability in all cities besides those in the top left corner (around Ho Chi Minh City). Climate change is anticipated to decrease flood damages costs in cities located at the bottom of the map (around Madrid), and decrease impacts on populations in cities in the mid-left (around Minneapolis) and mid-lower (again around Madrid) portions of the map. Socioeconomic development is projected to be the main driver increasing flood damages costs in cities on the upper-left triangle of the map (roughly from Mumbai down to Tebessa). Only in Ho Chi Minh City is development anticipated to be almost completely responsible for all increases in river flood impacts, all other cities in this study are at least partially affected by climate change. Development is not projected to play any part in a decrease in flood damages costs in any cities in this study (Caracas and Tebessa have no change in damages costs on the upper map, though it is attributed to development on the lower map).

Geographic regions are shown on Figure 2(b) with coloured text backgrounds. Cities in Southeast Asia are mostly found at the top of the map indicating high projected increases in overall flood impacts. South Asian cities are mostly located in the two areas of the map with patterns of very high increases in flood impacts, split between those projected to be most affected by development (around Mumbai, top middle) and by climate change (around Puna, mid right). Many North African cities are located in the lower left, indicating anticipated reductions in flooding due to socioeconomic development. North American cities are spread across the middle of the map indicating a wide range of projected changes.

1    a)

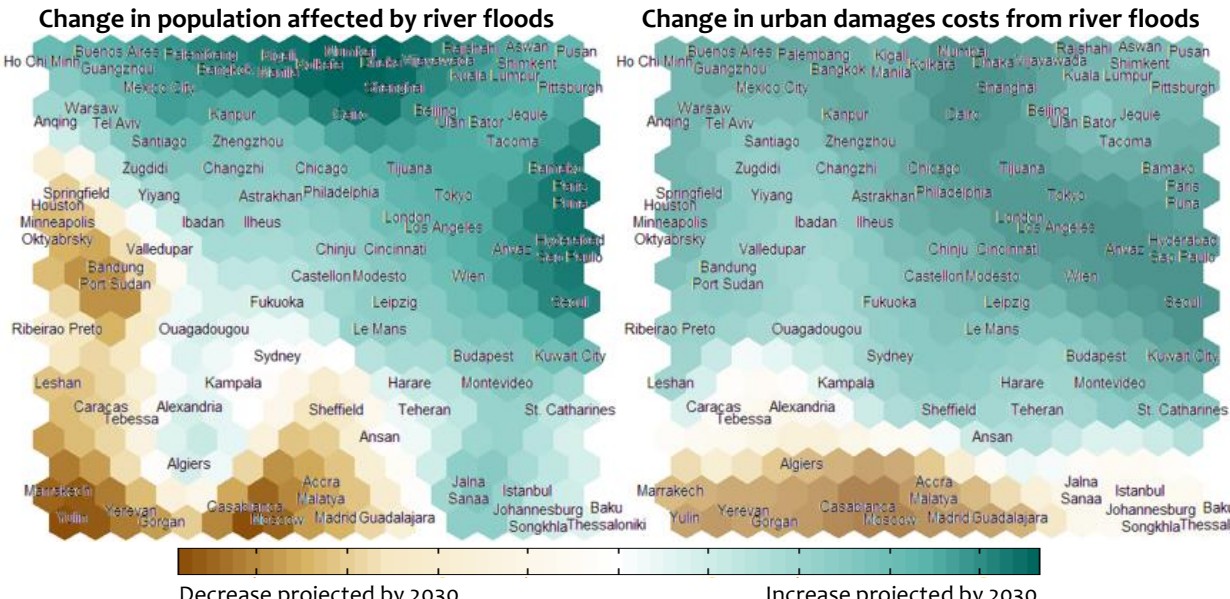

**Change in population affected by river floods**

**Change in urban damages costs from river floods**

Decrease projected by 2030                    Increase projected by 2030

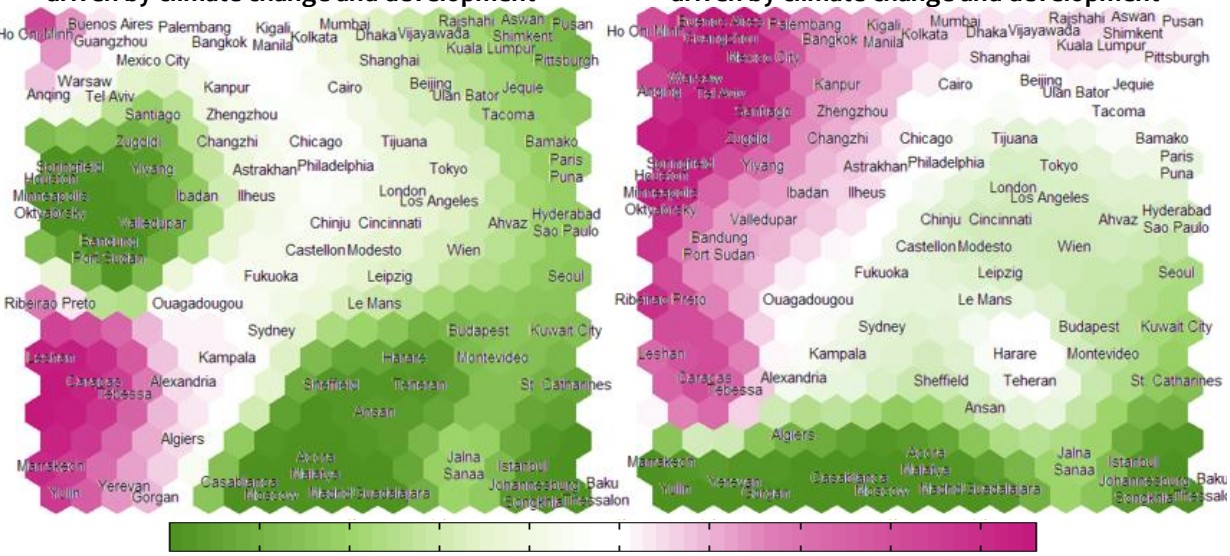

**Proportion of change in affected population driven by climate change and development**

**Proportion of change in damages costs driven by climate change and development**

More influence by climate change          More influenced by development

b)

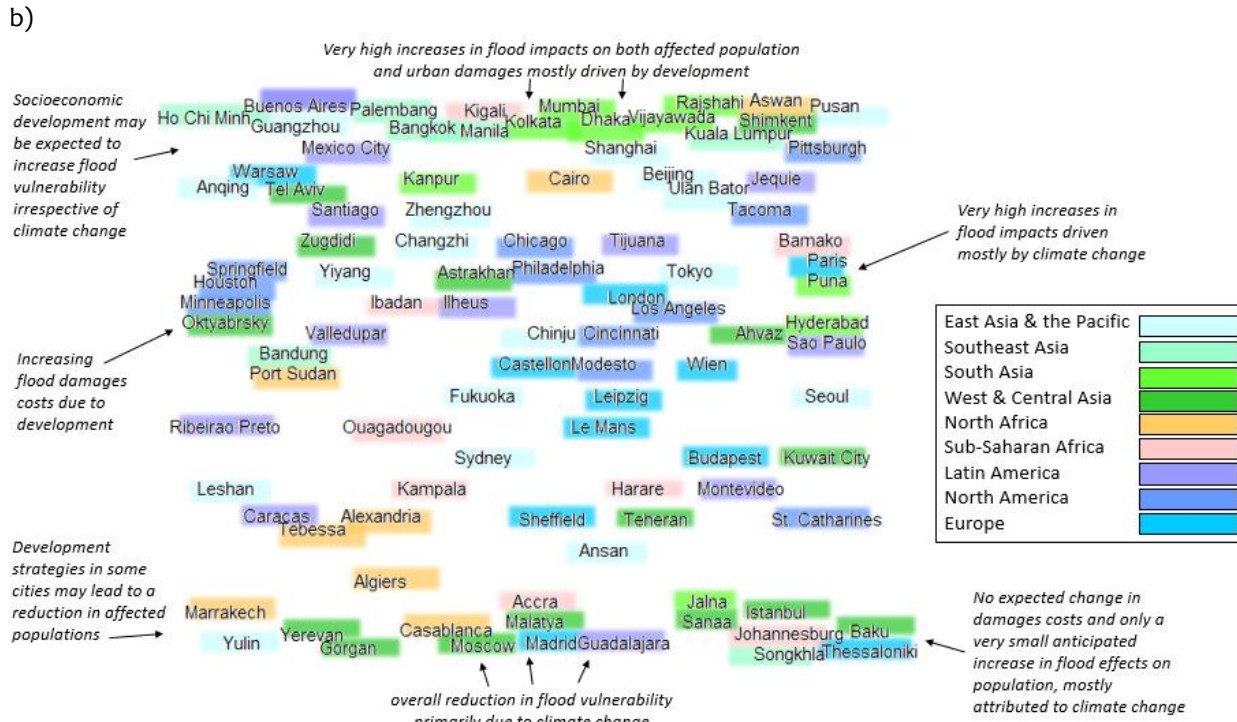

**FIGURE 2: SOM2 - PROJECTED CHANGES IN RIVER FLOOD IMPACTS WITH ASSOCIATED DRIVERS.**
River flooding in individual cities will be affected separately by climate change and development
between 2010 and 2030. Cities that are anticipated to experience similar pressures and responses in
terms of river flooding impacts are located nearby on the map. a) City labels are placed over coloured
copies of the map showing the relative values of each variable. b) City labels are coloured by region,
and characteristic patterns of general areas of the map are annotated.  The reader may refer to the
online version to zoom in on text if required.
Climate change and development may lead to opposing changes in a city's flood impacts on population and
property. A number of cities in this data set are predicted to have affected populations decreasing due to climate
change, whilst damages costs increase due to socioeconomic factors (around Springfield and Port Sudan, in the
mid-left). A decrease in flood effects on urban damages due to climate change, but an increase in affected
population largely due to development is, out of the cities in this study, only projected for Algiers (in the lower
left portion of the map).
In some cities, both drivers may generate changes in the same direction. For instance, in Marrakech, Yulin,
Yerevan and Gorgan, climate change is projected to be responsible for a decrease in damages costs whilst
socioeconomic development is anticipated to play a major role in the decrease in affected population, suggesting
that the reduction of population vulnerability due to development is complementing the direction of change
instigated by climate change. In certain cities near the upper left of the map (Santiago, Zugdidi and Yiyang), an
overall increase in flood impacts is expected, with increases in affected population almost completely attributed
to climate change and increases in damages costs almost completely attributed to development.

# SOM3: TEMPORAL PATTERNS

Relationships between the baseline characteristics and projected future changes of urban flooding in the individual cities are shown in Figures 1 and 2 respectively, however potentially similar temporal patterns between the cities are not evident from these maps. To link the information abstracted from the first two maps, we create a temporal map, SOM3, shown in Figure 3. SOM3 identifies which cities experience similar baseline flooding, are expected to incur comparable future hydrologic pressures from climate change and/or development, and are projected to respond in similar ways (or which cities may diverge in the future from similar baseline conditions).

The nodes of SOM3 are coloured based on the results of a second-level clustering, giving a visual separation to clusters of more similar data. Clusters are numbered from 1 to 16 for reference. As the cities have been located on SOM3 based on their locations on the 'baseline' and 'future projected changes' SOMs (in which the values of the variables vary smoothly though not monotonically along the axes), again the characteristics of the cities will flow smoothly along the map though multiple peaks and troughs of each variable are possible. The gradients of the cluster characteristics are indicated along the axes in Figure 3(a), which are nonlinear in data space.

Broad overviews of the patterns represented by certain regions of the map are identified on Figure 3 with arrows. The largest increases in flood effects are generally represented by nodes in the lower half of the map, whilst the largest decreases in flood effects are represented by nodes in the top left. Climate change is predicted to be the main driver of changes in population vulnerability along the top and down the left and right sides of the map, and in urban damages on the top and right of the map; therefore, climate change is the leading driver of changes in flood impacts on both population and damages costs at the top of the map. Development is the main driver of changes in flood impacts on populations in the lower and upper left side of the map, and on urban damages in the lower left area of the map; therefore, development is the leading driver of changes in flood impacts on both population and damages costs in cities on the lower left side of the map.

On Figure 3(b) the city labels are coloured by geographic region allowing for a regional visual comparison. We see the cities of each geographical region are more spread out on SOM3 than on SOM1 where each region was generally contained in one or two broad areas of the map. For example, on SOM3 Cairo and Aswan are noticeably separated from other North African cities which are located close together. Although the cities of this region have differing baseline flood levels (as shown on SOM1), most are projected to incur some reduction in future flood impacts (as shown on SOM2), with the exception of Cairo and Aswan. These cities both have forecasts of increased flood impacts - for Aswan increased impacts on the population due to climate change and impacts on property due to development, and for Cairo future impacts are projected to increase due to a relatively even mixture of both drivers. Another example can be seen in the cities of the USA shown on the map. They have similar starting conditions, yet are in two well-separated clusters on SOM3 - those around Houston and those around Los Angeles. The cities clustered around Houston are characterised by low impacts on population but high damages costs projected to elevate due to development, implying the possibility for local improvements through planning or mitigation strategies. The cities clustered around Los Angeles, however, are characterised by high overall impacts projected to increase predominantly due to climate change. In Sub-Saharan Africa we see Kigali and Bamako (which have similar medium-high baseline flooding conditions) are both expected to see increased impacts, but the cities are separated by SOM3 as these flood increases are attributed to development in Kigali and climate change in Bamako.

1     a)

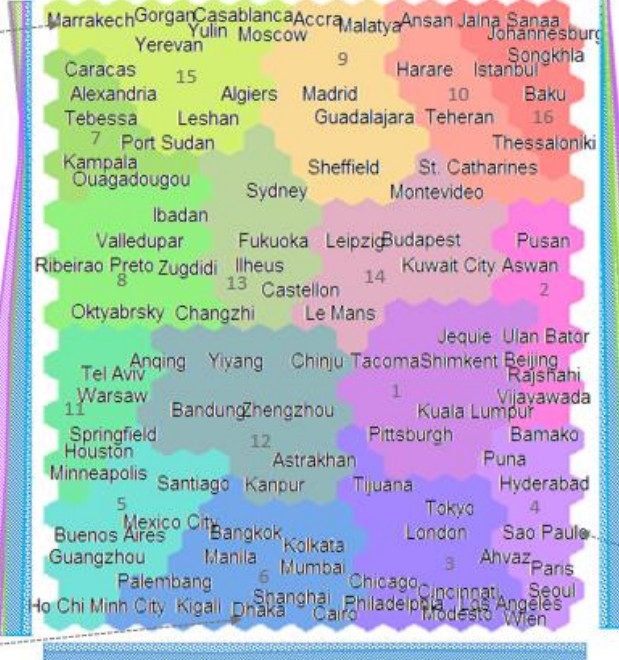

Greatest reduction in overall flood vulnerability

Lowest baseline flood impacts, with very small projected increases due only to climate change

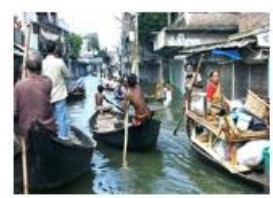

**Marrakech:** medium baseline flooding decreasing due to development and climate change

High damages costs with large projected increases due to socio-economic factors;

Very little or no population affected

Most increase in population affected due to climate change

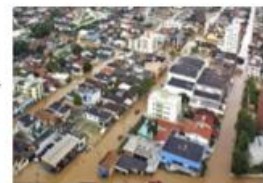

**Sao Paulo:** large increase in flooding due primarily to climate change

**Dhaka:** large increase in flooding due primarily to development

Largest baseline flooding, and highest projected increase in overall flood vulnerability (on left primarily due to development, on right due to climate change, and in the middle due to a relatively even mix)

| | |
|---|---|
| Baseline annual population affected | |
| Baseline annual damages costs | |
| Change in pop due to climate change | |
| Change in damages due to climate change | |
| Change in pop due to development | |
| Change in damages due to development | |

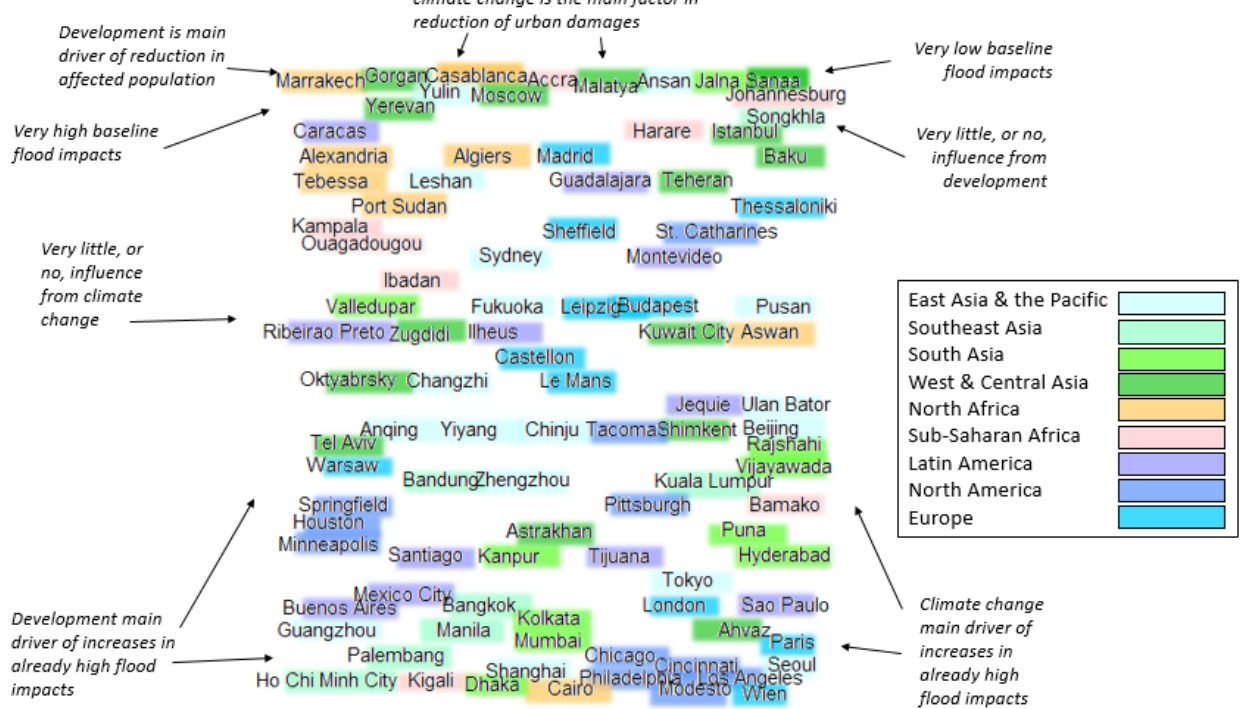

FIGURE 3: SOM3 - TEMPORAL PATTERNS – Cities are clustered close together that share similar
baseline (2010) flood vulnerabilities as well as similar anticipated changes driven by climate change
and development on population and urban damages costs by 2030. a) Locations of the cities are based
on their individual relationships to the principal curves in the baseline and future projected changes
data subsets - therefore, the axes represent the most important nonlinear gradients of flood
vulnerabilities in the data set. Coloured bars along the axes indicate the average levels of each variable
around the edges of the map. Cities are grouped into coloured clusters based on similarities. b) City
labels are coloured by region, and characteristic patterns of general areas of the map are annotated.
The reader may refer to the online version to zoom in on text if required.

To further analyse the characteristics of each cluster and the patterns found on SOM3, the properties of each
city in the 16 clusters are shown in a radial plot in Figure 4. Baseline values of population affected (blue, units =
number of people) and damages (orange, units = $US) are shown on a symmetrical logarithmic scale ranging
from -8 (ie. signifying a value of -100,000,000) to 11 (100,000,000,000) with the region between -1 and 1 on the
plot set as linear to avoid logarithmic discontinuities in the vicinity of zero. Zero is indicated by a dashed
circumference, and each progressive ring is an exponentially higher (or lower) value. Changes in population
affected and damages costs are shown on the same scale, in grey and yellow respectively. Values inside the
dashed (zero) circle represent decreases in flood impacts, and values outside represent increases, with the size
of the increase or decrease indicated by the distance from the dashed circle. The influence of climate change is
shown (light green for population and dark green for damages) on a linear scale from the same zero
circumference, in units of 'percentage of projected change attributable to climate change' (each progressive
ring is 10%). Green lines closer to the outer ring than the centre therefore indicate that the flood impacts on the
city are anticipated to be more influenced by climate change than by development. If the green lines are both in
the middle of the segment, this indicates a relatively equal influence of both drivers on both population and

property. Diverging green lines indicate that either population or damages costs are more influenced by climate
change, and the other by development.
From Figure 4, we can see the differences between neighbouring clusters, such as 10 and 16 located in the top
right of the map. Both clusters are characterized by low baseline impacts of flooding on the population, with
small increases in population impacts projected primarily due to climate change. However, cities in cluster 16
incur no flood damages costs at all in the baseline or future cases, yet in cluster 10 damages costs are projected
to increase due to climate change and development. Therefore, development has little or no impact on cities in
cluster 16 but does play a role in the increase in damages in cluster 10. In the top left of SOM3, we can now also
discern the difference between clusters 9 and 15. In both clusters, development is projected to have no impact
on the reduction of flood damages costs in most cities. Development does however play a strong role in the
reduction of flood impacts on populations in cluster 15 (except for Moscow and Casablanca) but none on
populations in cluster 9.

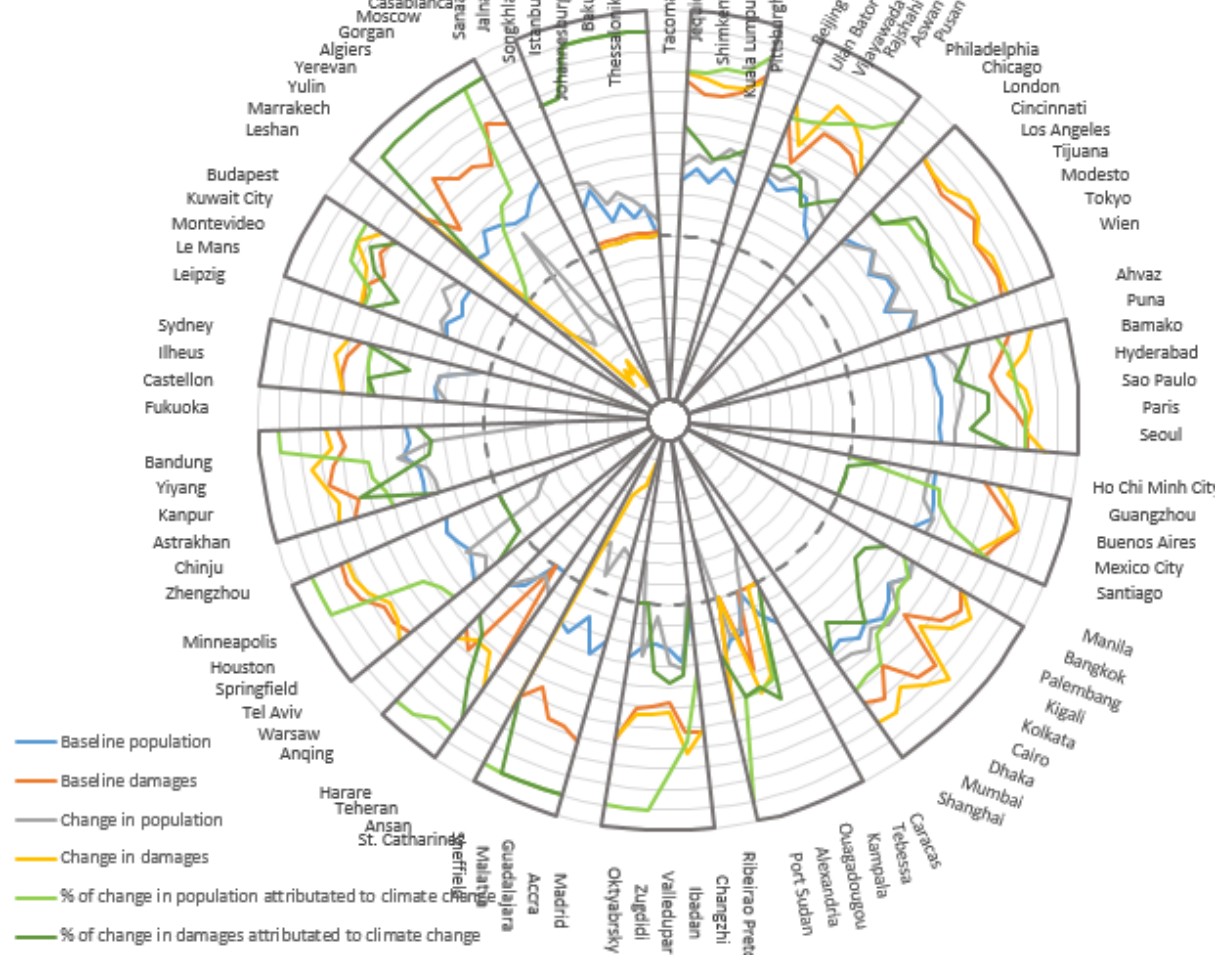

14          **FIGURE 4: RADIAL PLOT OF CLUSTERS OF FIGURE 3 –** The city members of the 16 clusters of
15          Figure 3 are shown with their individual variable values. The scale is logarithmic for baseline

1       and changes in population and damages, and linear for the percent of change attributed to
2       climate change, with the dashed circle representing zero.

The relationship between the two drivers, climate change and development, can be discerned from Figures 3
and 4. Climate change is projected to impact populations more than urban damages costs in clusters stretched
across the centre of the map (clusters 8, 11, 5, 6, 12, 14, 1, 10, 1, 2, and 4 - in cities in these clusters, the proportion
of change in the population affected attributed to climate change is higher than the proportion of change in
damages costs attributed to climate change). In cluster 15, the population is projected to be more influenced by
development than damages costs will be (a higher proportion of the change in damages costs is attributed to
climate change than for population). In the remaining clusters, climate change (and development) are projected
to affect the population and damages costs relatively similarly (clusters 7, 13, 9, 16 and 3). Some examples of
diverging impacts on population and damages costs stand out on the radial plot in Figure 4. For instance, in Port
Sudan, Sheffield and Bandung, significant reductions in affected population are projected to be 100% due to
climate change, however large projected increases (~300 to 400%) in damages are due mostly to development.
In Leshan, development is projected to slightly lower the amount of affected population and also to increase
damages costs more than three-fold.
## 4 DISCUSSION
In this study, the 'patterns' and 'clusters' in the data have been identified. The patterns, depicting unique, key
combinations of variables that are characteristic of the data set, have been extracted at three separate levels on
SOM1, SOM2 and SOM3 (for example, each pattern of SOM3 is a separate combination of levels of baseline and
projected future flood conditions as well as projected influences of climate change and development). The
clusters consist of groups of cities whose conditions are anticipated to be similar to these patterns, based on the
given data. In the previous section, the interpretation of each SOM (1-3) has been discussed. In this section, a
discussion of a selection of the patterns and clusters is provided. The information that can be gleaned from the
SOMs is related to information in current literature through the use of specific examples.
The lower left region of SOM3 highlights the fact that a number of cities already experiencing large flood effects
are anticipated to incur great flood increases influenced predominantly by socioeconomic factors. In these cities,
climate change is playing a large role, and yet it is overshadowed by the magnitude of regional economic growth
(UNEP, 2016; website 8) leading to migration, changing land use and unplanned development in flood zones.
Many of these cities are in Asia, where the climate is experiencing warming trends, increasing temperature and
precipitation extremes, and rapid glacial melting resulting from climate change (Pachauri et al., (2014) chapter
24: 'Asia'). However, it can be seen that socioeconomic growth in this area is projected to have even more of an
impact on urban floods than climate change.
Pachauri et al. (2014, chapter 24: 'Asia') assert that human and material losses due to flooding are already heavily
concentrated in India, Bangladesh and China and Jongman (2012) estimates the largest current and future
economic exposure to river floods to be in Asia. As an example, we take a closer look at Dhaka which, with a GDP
per capita of $1212 in 2015, already has one of the highest levels of population affected annually by flooding (over
130,000) and this number is projected to increase almost five-fold (to over 630,000) by 2030. The greatest
change predicted for Dhaka, though, is an almost 22-fold increase in annual damage costs (from $8 million to
$175 million). Dhaka is subjected to regular flooding from surrounding rivers, with peak flows in the Brahmaputra
and Ganges Rivers coinciding to exacerbate flood impacts. In the past, most low-lying areas of western Dhaka
were infilled for residential and commercial use, causing a reduction in areas for flood water storage.
Uncontrolled urban expansion is spreading rapidly across the floodplains in the east of the city placing more

people in flood hazard zones (Kreimer et al., 2003). These hasty developmental changes are having more of an impact on the urban hydrology of Dhaka than climate change is. Examples of cities in similar situations include: Kolkata (with the highest baseline affected population in this study), Mumbai (with a seven-fold increase in both population affected and damages due 40% and 60%, respectively, to development), Bangkok (with large increases, 50-75% of which are attributed to development) and Ho Chi Minh City (with a 50% increase in affected population and an over five-fold increase in damages costs, almost entirely attributed to development).

On SOM3 we can identify that developmental changes in some cities appear to be effectively reducing impacts from river flooding, for example in Marrakech (cluster 15). The affected population level is projected to decrease mostly due to socioeconomic factors (website 3; Ward et al., 2013; Winsemius et al., 2013). Through an 'Integrated Disaster Risk Management and Resilience Program for Morocco' (World Bank, April 2016-Dec 2021), Morocco is becoming more resilient to climate change and less vulnerable to natural hazards, and is ensuring a rapid transition to a low-carbon economy. Through Morocco's National Strategy for Sustainable Development, a commitment has been made to reduce national greenhouse gas emissions by 32% by 2030. This will be done through an increase in renewable energy sources to 50%, a reduction in energy consumption by 15%, as well as various agricultural, water, waste, forest, industry and housing initiatives (website 9). These housing initiatives in Marrakech include a slum clearance and relocation project, which has become part of urban policy (Ibrahim, 2016), reducing the amount of people inhabiting flood hazard zones. Alert systems in the valleys of the Atlas region above Marrakech have been improved, and the proportion of the population living in slums has decreased from over 8% in 2004 to less than 4% in 2010 (UN-Habitat website). The urbanization rate in Morocco is also projected to slow down towards 2030 (UN-Habitat website). This risk-prevention approach combining early warning systems, relocation of inhabitants out of the flood zone, and less urban expansion is expected to combine to reduce the impact of floods on the population of Marrakesh.

Current high flood impact conditions projected to get much greater primarily due to climate change are anticipated for cities in the lower right of SOM3, with high magnitude changes expected for impacts on both population and property. One of these cities, Sao Paulo, is expected to experience an almost seven-fold increase in both the number of population affected (to over 140,000 annually) and urban damages costs (to over $500,000,000 annually) by 2030. 15% of the change in population and 35% of the change in damages is attributed to development, but the majority of the change is projected to come from climate change. Sao Paulo, the largest city in Brazil, has a city footprint projected to increase over 38% by 2030, by which time 22% of the urban area may be located in flood zones (Young, 2013). The IPCC (Pachauri et al., (2014) chapter 14 'Latin America') predicts the increase in temperature in central and south Brazil to be the largest projected increase in Latin America, which will be combined with a 10-15% increase in autumnal precipitation, greatly affecting the hydrologic cycle in the region. The substantial change in development is therefore expected to be eclipsed by the even greater projected change in climate in Sao Paulo, and other cities in this region of SOM3.

The anticipated reduction in flood damage costs caused by climate change evident in Cluster 15 may be a result of changing snow melt conditions upstream of these cities. It has been shown that some global regions will experience a decreasing trend in the magnitude and frequency of snow melt floods as the climate warms, as well as a shift in the timing of these floods (Schiermeier, 2011; Barnett et al., 2005; Immerzeel et al., 2010).

Many high-income cities with already high current flood vulnerabilities have projections for large elevations in damage costs, but not increased levels of affected population. This can be seen in cities on SOM3 centred around London, Tokyo, LA and Vienna (cluster 3), and Sydney and Castellon (cluster 13). Through high levels of planning, preparedness and infrastructure, prosperous regions generally have systems in place to minimize flood impacts on the population, even though they may incur large economic losses (Desai et al., 2015; Kreimer et al., 2003).

Almost half of the projected increases in these clusters are attributed to development, suggesting that these cities may have the capacity for lessening potentially elevated flood damage costs by concentrating on planning and mitigation policies.

Although changing climate in some areas is projected to lessen regional flooding, development within urban flood zones may be severe enough to offset any reductions in flood impacts. This can be seen most prominently in a strip on the left of SOM3 stretching from Port Sudan down to Santiago.

Though this study does not consider coastal flooding, it may be noted that due to their locations near river mouths, many of the cities in the lower left of the map that are projected to experience high increases in impacts from river flooding are also at risk of increased coastal flooding from intensified storms and sea level rise due to climate change. Mumbai, Guangzhou, Shanghai, Ho Chi Minh City, Kolkata, Bangkok, and Dhaka are 7 of the top 14 cities (out of 136) ranked by current population exposure to coastal flooding. These same cities also comprise the top 7 cities (in this order: Kolkata, Dhaka, Mumbai, Guangzhou, Ho Chi Minh City, Shanghai, Bangkok) ranked by future (2070) estimated population exposed to coastal flooding (UNEP, 2016; Nicholls et al., 2008).

Almost all projected changes in flooding in this data set are of a relatively similar order of magnitude to the original effects, as can be observed on Figure 4. That is, most cities that are only marginally affected by flooding in 2010 are projected to experience only small increases by 2030, whereas cities with larger flood effects can expect greater changes indicated by the significant correlation between the magnitudes of the cities' baseline flooding effects and the changes projected by 2030 (log-transformed absolute values for both variables) – an 88% correlation exists in the number of population affected and a 94% correlation for property damage costs. This supports the findings of Milly et al. (2002) who observed that the frequency of large flood events in large basins had increased substantially in the 20[th] century, but smaller floods had not.

The analysis in this paper is based solely on the data provided in Aqueduct, regardless of the extent to which on-the-ground flood management measures are incorporated into the socioeconomic models which produced this data. A discussion characterizing individual cities is included here as a point of interest to relate the data to current national conditions, providing possible reasons why these cities may fit into the map where they do.

## 5 CONCLUSION

This study adds to the understanding of natural hazards in a global context, an important aspect of regional disaster risk management due to the dependency of local situations on global processes (Desai et al., 2015). Complex, nonlinear social-environmental relationships make it difficult to anticipate local responses to global changes (Desai et al., 2015), and this study contributes to risk communication (the process between risk perception and adaptation planning (Cardona et al., 2012)) providing a visual analysis of global patterns of evolving flood impacts, socioeconomic development and climate change, and the local city-level consequences of these changes.

Global patterns of urban flood responses to global and local changes in hydrology driven by climate change and development have been identified and visually communicated through a series of related self-organizing maps. Cities have been matched to these global patterns, and relationships between the individual cities have been discerned with respect to baseline flooding conditions and expected future changes. The visual analysis in this study has revealed interesting city-level patterns that are otherwise unobservable in the complex data set, and provides a comparison and distinction between individual cities that is not apparent in regional- or economic-level projections.

We have performed dimension reduction and clustering with a series of self-organizing maps to identify changing global patterns of city-level flood risks. The maps provide an indication of the predominant characteristics which determine the differences in urban river flood impacts between cities, and the cities occupy positions on the maps signifying their relative conditions. The method used here incorporates adaptions to the self-organising map technique for map shape selection and temporal pattern extraction, allowing two levels of information to emerge: the characteristic patterns of dynamic global urban flood vulnerabilities, and a comparison between the cities with respect to flood characteristics and trends. This SOM method could be adopted for visualising and clustering any large data set in which the underlying intervariable relationships are difficult to explicitly define, such as is common with human-environmental interactions, and which change over time. The resulting visualisations produce an overall impression of the temporal structure and clusters in the data, allowing for a readily understood overview of the relationships between the variables, and amongst individual data items.

A shortcoming of the method used in this study is the assignment of flood protection level based on an assumption of proportionality with national income level. As standardised, current information on the real flood protection levels of all the cities in the data set is not readily available, this assumption has been necessary and has been made in line with current practice. This limitation has been recently acknowledged in the literature, with Winsemius et al. (2016) noting that 'currently installed flood protection is an important missing link in the assessment of global flood risk'. Future studies may aim to include specific flood protection levels for each city.

Whilst the timeline of this study is short, it is restricted by the data that is available. Studies at a global scale have been traditionally limited due to lack of cohesive data sets, and therefore the data set provided by Aqueduct is valuable for the fact that it spans a global set of cities and provides a rare opportunity for comparison. As the data is only provided for 2010 and 2030, there was no prospect for a longer analysis. Whilst this analysis may not provide a long-term outlook, at the very least an important insight into current and near-future conditions can be gained.

Cities have major implications for climate change mitigation and adaptation (Revi et al., 2014). Unplanned development and urban migration are increasing vulnerabilities to natural hazards (UNEP, 2016) and land cover change and greenhouse gas emissions are intensifying urban hydrology. Understanding the relationship between flood impacts and social vulnerability is a necessary step for prioritizing flood mitigation and prevention strategies (Doocy et al., 2013). Whether the main driver of increased urban flood impacts is development or climate change, cities will benefit from development restrictions and planning standards for urban expansion, sustainable land development, management of population distribution and migration, and early warning systems and preparedness (Revi et al., 2014; UN-DESA, 2014; Doocy et al., 2013).

Future work may include the addition of greenhouse gas emissions data, geographic location, city sizes and densities to this study, to discern the relationships of these factors with urban flood changes. Greenhouse gas emissions are the largest contributor to global warming, leading to alterations in the intensity of the hydrologic cycle (Pachauri et al., 2014, Barnett et al., 2005; Wentz et al., 2007; Schiermeier, 2011), and cities are the major contributors of greenhouse gases, with a large proportion of global emissions produced by a small global land area (Mills, 2007; Angel et al., 2010; Revi et al., 2014). The addition of these elements could highlight the essential role cities could play in climate change mitigation and the reduction of urban flood impacts.

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

## 2   7 Competing interests

3   The authors declare that they have no conflicts of interest