# Peer review of "Patterns and comparisons of human-induced changes in river flood impacts in cities"

_Hydrology and Earth System Sciences, 2017_

## Referee Comment (RC1) · Anonymous Referee #1 · 12 Jul 2017

The paper explores the global dataset on fluvial flood risk prepared and discussed by Philip Ward and co-authors on city level. This is potentially a good idea; although the data has been explored in a series of papers by these authors there are still room for further studies. In its scope the current paper is rather close to Winsemius et al (2016); the current paper focuses on cities while the original paper focuses on river catchments.

There are however a number of important shortcommings that should be addressed before I potentially can recommend publication.

First of all I would like to see a discussion of what new knowledge the authors think they can gain by considering cities rather than river catchments. At least the findings of the current paper should be compared to Winsemius et al (2016).

[Figure]

Next, the introduction is rather long and gives a thorough introduction to flooding. Unfortunately the authors does not distinguish between the different types of flooding that can occur, i.e. sea surges, groundwater induced flooding, pluvial flooding and, fluvial flooding. The paper is based on a specific data set that only considers fluvial flooding. Hence references to authors that specifically refer to pluvial flooding should be removed, e.g. Willems et al (2012). Perhaps differences in fluvial flood risk between cities can be explained by different exposures to other types of flood risk? If the authors do not wish to enter such discussions they should stick to considering only one type of flooding. Next, the literature on whether flood risk is stationary or has an increasing trend is quite abundant. The findings differ, primarily as a function of the framing, i.e. if the models include corrections for changes in socio-economic development, vulnerability, etc. The author uses the terms 'impacts', 'risks' and 'material damage' more or less inter-changeably throughout the paper. When referring to IDSR and other recognized frameworks I would expect more stringent use of the terminology and more transparent explanations and assumptions. This includes a description of what the scenario for development is between 2010 and 2030 and the rationale for choosing the approximately 100 cities in Table 1, that are then later reduced to 80 cities (without mentioning which of the cities are excluded).

However, my most important concern is that I cannot see what the authors are doing with the data. The closest to an explanation is that the authors state that the calculations are carried out in Matlab using a modified version of the package SOM. Modified in which way? Why are the data transformed the way they are if the SOM approach is particularly good in dealing with non-linear relationships, how do you treat taking logarithm of the value of zero, what are the definitions of QE, TE, DRR, and the Davies-Bouldin Index, and how is a U-matrix visually verified? Without this information the reader will have to do a complete reanalysis to (perhaps) get the same results as the authors.

Since I expect a thorough revision of the method section before perhaps resubmitting

the paper I have not read the results and discussions sections in detail.

More detailed comments: I disagree with the use of the term 'spatio-temporal'. It usually denotes something where there is explicit reference to a spatial dimension, in the current example perhaps the physical distance between the cities. I think the authors should find another term to describe the characteristics of their data.

The data on the figures cannot be read in the pdf-version of the paper.

I cannot follow the discussion on the cluster. Perhaps it is just me not being able to see the same patterns as the authors.

The list of references should be improved. Just from browsing it I can see dubious referencing to e.g. IPCC (2014) (Use author names) and Willems (2012) (several authors), and Kohonen (2001) (incomplete reference). There are more errors than the ones I have pointed out.

---

## Referee Comment (RC2) · R. Larsson (Referee) · 9 Aug 2017

General comments The topic of this paper is clearly both interesting and of great importance. The paper is well written and has a clear structure. However, it seems to me that the paper does not really fulfil the promises implicit in the title. Which are the patterns in flood impacts on cities that are revealed in the paper?

There are some problems with the method used, and there is a problem with the general approach. The latter problem is related to the presumption that it is possible to draw conclusions about individual cities based on a global model. It is also doubtful whether any meaningful patterns can emerge from such a rather superficial study. The method issues are covered below.

[Figure]

Before publication, technical corrections need to be made. Specific comments can be dealt with by comments/clarifications in the paper or via rebuttal.

Specific comments (p.3, lines 41-44) The objective is given only in rather general terms. It would have helped the reader to get some more precise information about which types of "global patterns and relationships" that is meant to emerge from the study.

(p.4, lines 11-20) The main problem with the method is the data source used in the paper. This source consists of output from a global tool produced by the World Resources Institute. In the paper this model is described as made up of "global hydrologic and hydraulic models" and more. It raises some concern about the accuracy of such models when they are "global". Unfortunately, the only reference in the paper leads to a web site, which in turn refers to the name of a model but with no proper literature reference. So, it is quite difficult for the reader to judge for herself how useful the data used in the paper is. Considering the spatial resolution given in the said website the usefulness is doubtful. As an example of interest for the reviewer: the country of Sweden is presented as one "basin" !

(p.4, lines 29-34) Another weakness in the method is that the actual, real flood protection level is not included. Instead flood protection level is based on the assumption of proportionality with national income level.

(p.4, line 18-22, 38) On the one hand: changes due to socioeconomic development are driven by population and economy. On the other hand: protection level, which is assumed proportional to national income level is kept constant over time. This is inconsistent.

(p. 6, lines 16-19) The present situation is in the paper characterized based on conditions 2010 while effects of climate change is based on projections for 2030. This time interval, 20 years, is a bit short for meaningful comparisons.

(p.8, 3 lines 5-15; Fig. 1a) While the colouring of the maps allows for different absolute

impact on people and material damage for an individual city, the location of all cities are exactly the same on both people and damage maps. This latter fact seems to indicate that the relative difference in impact between cities remains almost identically the same for both people and damage.

(p.17, lines (23-38) The actual, on the ground flood management measures and other socioeconomic development are presented for Marrakech. However, in the model producing the source data for this paper only some of these factors were incorporated. See previous comment re page 4.

Technical corrections (p.1) The title is ". . ...changes on river. . ." should be ". . ...changes in river. . ." (p.7: 25) ". . ...Whist. . ." should be ". . ...Whilst . . ." (Fig. 1) Fig. 1a) is difficult to read because of the many cities and the small font; however still ok. Fig. 1b) is too blurry due to combination of background colours and text. More or less impossible printed; still difficult on the screen. (Fig.2) Similar problems as with Fig. 1 (Fig.3) Similar problems as with Fig. 1 &2. Difficulties reading this figure exacerbated by all the information crammed into one page. The gradients indicated along the axes are impossible to interpret. (p.13:1) ". . ...reduction on. . ." should be ". . ...reduction of . . ." P.16:16) ". . ...loses. . ." should be ". . ...losses . . ."

---

## Editor Comment (EC1) · E. Toth (Editor) · 9 Aug 2017

Dear Authors, while finding the topic and the approach interesting, both referees have raised serious concerns regarding both the data (Ref#2) and the method (Ref#1), and I agree that more information on both are therefore certainly needed, in addition to highlighting the need to clarify better the final objectives (and how to 'measure' them) and the value and transferability of the work. The Reviewers have uploaded their comments in advance of the discussion end and I would therefore ask you to provide your replies (two different Authors' Replies, one for each referee comment) as soon as possible, so that the Referees may, if they want, comment back on your replies before the end of the discussion phase. I look forward to reading your replies. My best wishes, Elena Toth HESS Topical Editor

---

## Author Comment (AC1) · 18 Aug 2017

We thank you for your time and effort in providing comments on this paper.

The paper explores the global dataset on fluvial flood risk prepared and discussed by Philip Ward and co-authors on city level. This is potentially a good idea; although the data has been explored in a series of papers by these authors there are still room for further studies. In its scope the current paper is rather close to Winsemius et al (2016); the current paper focuses on cities while the original paper focuses on river catchments.

As mentioned, Winsemius et al (2016) analyses the data set with respect to river basin and income level. Our analysis is based on city-level flood impacts. Whilst similar, these studies are not the same, and the insights revealed in this paper do not overlap with those provided in Winsemius et al (2016).

There are however a number of important shortcommings that should be addressed before I potentially can recommend publication.

First of all I would like to see a discussion of what new knowledge the authors think they can gain by considering cities rather than river catchments. At least the findings of the current paper should be compared to Winsemius et al (2016).

The comparison of cities allows an understanding of conditions that can be expected on a city-level, and identification of cities facing similar circumstances. As climate change, development, and urban administrations transcend river basin boundaries, it is sensible to investigate impacts and determine potential mitigation strategies at the city level (as well of course as basin level, but that has been covered in Winsemius et al (2016)).

This is discussed in the conclusion of this paper, p19 lines 12-26, which is reproduced here:

'This study adds to the understanding of natural hazards in a global context, which is an important aspect of regional disaster risk management due to the dependency of local situations on global processes (UNISDR, 2015). The complex nonlinear socio-environmental relationships make it difficult to foresee local responses to global changes (UNISDR, 2015), and therefore this study focuses on risk communication (the process between risk perception and adaptation planning (Cardona et al., 2012)) to provide a visual analysis of the global patterns of evolving flood impacts, socioeconomic development and climate change, and the local city-level consequences of these changes.

Cities have major implications for climate change mitigation and adaptation (Revi et al., 2014). Unplanned development and urban migration are increasing vulnerabilities to natural hazards (UNEP, 2016) and land cover change and greenhouse gas emissions are intensifying urban hydrology. Understanding the relationship between flood impacts and social vulnerability is a necessary step for prioritizing flood mitigation and prevention strategies (Doocy et al., 2013). Whether the main driver of increased urban flood impacts is development or climate change, cities will benefit from development restrictions and planning standards for urban expansion, sustainable land development, management of population distribution and migration, and early warning systems and preparedness (Revi et al., 2014; UN-DESA, 2014; Doocy et al., 2013).'

Next, the introduction is rather long and gives a thorough introduction to flooding. Unfortunately the authors does not distinguish between the different types of flooding that can occur, i.e. sea surges, groundwater induced flooding, pluvial flooding and, fluvial flooding.

It is stated on p4, lines 13-14, that this paper only considers fluvial flooding:

> 'This data is solely related to the influence of fluvial flooding on metropolitan areas, and does not include coastal or pluvial flooding.'

If this statement would be better placed in the introduction, it can be moved, however we are conscious that the introduction is already 'rather long'.

The paper is based on a specific data set that only considers fluvial flooding. Hence references to authors that specifically refer to pluvial flooding should be removed, e.g. Willems et al (2012).

We refer to Willems a few times, for example in the following statement (p2 lines 36-37):

> 'Increases in rainfall intensity at urban hydrology scales of up to 60% are anticipated by 2100 (Willems, 2012).'

Though Willems may be generally interested in pluvial flooding, any references made here concern Willems' statements on increases in rainfall, not flooding.

Perhaps differences in fluvial flood risk between cities can be explained by different exposures to other types of flood risk? If the authors do not wish to enter such discussions they should stick to considering only one type of flooding.

The comment 'if the authors do not wish to enter such discussions they should stick to considering only one type of flooding' is confusing for three reasons.

First, we have made it clear we are only considering one type of flooding - fluvial flooding (see previous comment).

Second, we have 'entered such discussions'. The exposures of various cities to other types of flood risk is discussed in the 'discussion' section of the paper (eg. p18 lines 22-28, as reproduced here). Unfortunately, the reviewer has stated (below) that they have not read this part of the paper.

> 'Though this study does not consider coastal flooding, it may be noted that due to their locations near river mouths, many of the cities in the lower left of the map that are projected to experience high increases in impacts from river flooding are also at risk of increased coastal flooding from intensified storms and sea level rise due to climate change. Mumbai, Guangzhou, Shanghai, Ho Chi Minh City, Kolkata, Bangkok, and Dhaka are 7 of the top 14 cities (out of 136) ranked by current population exposure to coastal flooding. These same cities also comprise the top 7 cities (in this order: Kolkata, Dhaka, Mumbai, Guangzhou, Ho Chi Minh City, Shanghai, Bangkok) ranked by future (2070) estimated population exposed to coastal flooding (UNEP, 2016; Nicholls et al., 2008).'

Third, the reviewer has implied in a previous comment that it is unfortunate different types of flooding are not distinguished: 'Unfortunately the authors does not distinguish between the different types of flooding that can occur, i.e. sea surges, groundwater induced flooding, pluvial flooding and, fluvial flooding.'

Therefore, it is unclear what the reviewer is expecting to result from this comment.

Next, the literature on whether flood risk is stationary or has an increasing trend is quite abundant. The findings differ, primarily as a function of the framing, i.e. if the models include corrections for changes in socio-economic development, vulnerability, etc. The author uses the terms 'impacts', 'risks' and 'material damage' more or less inter-changeably throughout the paper. When referring to IDSR and other recognized frameworks I would expect more stringent use of the terminology and more transparent explanations and assumptions.

The terms 'impacts', 'risks' and 'material damage' will be defined and their use updated.

This includes a description of what the scenario for development is between 2010 and 2030 and the rationale for choosing the approximately 100 cities in Table 1, that are then later reduced to 80 cities (without mentioning which of the cities are excluded).

The development scenarios are discussed in the Data section (p4 lines 18-28) with references to the models that were used to produce the data and accompanying documentation. The specific scenario is not described explicitly, however enough detail is provided for an interested reader to find the specific details:

> 'Three separate scenarios of climate change and socioeconomic development (optimistic, business-as-usual, and pessimistic) are given in Aqueduct, and in this study we use data from the business-as-usual case for our future flood impact scenario.'

The rationale for choosing the cities is explicitly given on p4, lines 6-8:

> 'The selection of cities used here is based on a list provided by the Lincoln Institute of Land Policy's Atlas of Urban Expansion (Angel et al., 2010, website 1), spanning all continents except Antarctica, encompassing four economic levels and four population levels.'

The reason that some of these cities have been excluded is explicitly state on p5, lines 5-6:

> 'Cities with no flood impacts in both 2010 and 2030 were removed (22 cities), though cities with no flood impacts in 2010 but with flood impacts in 2030 have been kept in the study.'

The reviewer is wondering why there is no mention of which cities were excluded - it was decided that a list of cities that are NOT included in this study would not be of interest to the readers hence they have not been listed.

However, my most important concern is that I cannot see what the authors are doing with the data. The closest to an explanation is that the authors state that the calculations are carried out in Matlab using a modified version of the package SOM. Modified in which way? Why are the data transformed the way they are if the SOM approach is particularly good in dealing with non-linear relationships,

The data is log transformed, following Agarwal and Skupin's (2008) recommendation that highly skewed variable distributions may benefit from log transformation before use in the SOM. *[Agarwal, P. and A. Skupin (2008). Self-organising maps: Applications in geographic information science, John Wiley & Sons.]* This information and reference will be added to the methods section.

how do you treat taking logarithm of the value of zero,

The treatment of the logarithm of values between -1 and 1 is carefully described on p15, lines 12-13.

what are the definitions of QE, TE, DRR, and the Davies-Bouldin Index,

QE, TE, and DRR are defined on p7, lines 16-19. A reference is provided for the Davies-Bouldin Index with the information that it is used to determine the number of clusters (p7, line 31).

and how is a U-matrix visually verified? Without this information the reader will have to do a complete reanalysis to (perhaps) get the same results as the authors.

Further SOM references have been provided for readers who are unfamiliar with the method, though it has been summarised in the Methods section. In an effort to avoid repetition of previously published material, the SOM method has not been reproduced in detail. As the papers and books

outlining the SOM method have already been cited over 40,000 times, it was not felt that this method needed further repetition. With any paper, it is hard to know how much background knowledge to expect the readers to possess, and therefore we provide direction to further reading for those unfamiliar with certain aspects of the background.

Since I expect a thorough revision of the method section before perhaps resubmitting the paper I have not read the results and discussions sections in detail.

We are disappointed to hear this. I expect the reviewer would have found some answers to their questions had they read the paper.

**More detailed comments:**

I disagree with the use of the term 'spatio-temporal'. It usually denotes something where there is explicit reference to a spatial dimension, in the current example perhaps the physical distance between the cities. I think the authors should find another term to describe the characteristics of their data.

The term 'spatiotemporal' is commonly used to refer to data in which there is a cross-sectional structure to the data as well as a temporal one. In this case these are: 1) the state of flooding at each timestep, 2) and the timesteps themselves. The spatial dimension does not always refer to geographic distance.

The data on the figures cannot be read in the pdf-version of the paper.

The figures could be reproduced with larger text.

I cannot follow the discussion on the cluster. Perhaps it is just me not being able to see the same patterns as the authors.

Which discussion on 'the cluster'?

The list of references should be improved. Just from browsing it I can see dubious referencing to e.g. IPCC (2014) (Use author names) and Willems (2012) (several authors), and Kohonen (2001) (incomplete reference). There are more errors than the ones I have pointed out.

The references will be checked and updated where necessary.

---

## Author Comment (AC2) · 18 Aug 2017

Thank you very much for your time in reviewing this paper, Dr Larsson. We appreciate all comments and will address each one as thoroughly as possible to produce an improved paper. It appears your main concern rests with the data source used in the paper. We understand that more information is needed in the referencing of this data set, and hope that the information provided below will allay these concerns.

**General comments:**

The topic of this paper is clearly both interesting and of great importance. The paper is well written and has a clear structure.

However, it seems to me that the paper does not really fulfil the promises implicit in the title. Which are the patterns in flood impacts on cities that are revealed in the paper?

The 'patterns' are the key characteristics extracted for each cluster of cities sharing similar baseline and projected flood conditions. Each city in each cluster is matched to the 'pattern' represented by the cluster centroid. As described on p6 lines 1-3, the values of the prototype vectors (map nodes) come to represent the prevalent patterns in the data after they have self-organized amongst the data items. For example, cities in cluster 10 in the upper right of Fig.3 share the 'pattern' of low baseline flooding impacts, with small increases in population impacts projected primarily due to climate change, and increases in material damages projected due to both climate change and development. For Fig3, the text on p13 and 15, and Fig 4 on p14 interpret these patterns in detail. The Method section (p6) may benefit from expansion, as it appears more detail is needed on the process of pattern extraction by the SOM.

There are some problems with the method used, and there is a problem with the general approach. The latter problem is related to the presumption that it is possible to draw conclusions about individual cities based on a global model. It is also doubtful whether any meaningful patterns can emerge from such a rather superficial study. The method issues are covered below.

It is not within the scope of our study to assess whether the model used to produce the data is comprehensive, but rather to provide a visualisation of the available data. The data set as published is immense and difficult to comprehend without a reduction and visualisation, which we are providing. Any conclusions about individual cities are based solely on the data items presented, and comparisons are based on similarities between data items, be them cities or merely numeric vectors.

**Specific comments:**

(p.3, lines41-44) The objective is given only in rather general terms. It would have helped the reader to get some more precise information about which types of "global patterns and relationships" that is meant to emerge from the study.

This sentence had previously been more long-winded, but was reduced to this phrase for brevity with the knowledge that the 'patterns and relationships' would be explained in detail throughout the paper. This sentence can certainly be expanded to detail the expected patterns and relationships.

(p.4, lines 11-20) The main problem with the method is the data source used in the paper. This source consists of output from a global tool produced by the World Resources Institute. In the paper this model is described as made up of "global hydrologic and hydraulic models" and more. It raises some concern about the accuracy of such models when they are "global". Unfortunately, the only reference in the paper leads to a web site, which in turn refers to the name of a model but with no proper literature reference. So, it is quite difficult for the reader to judge for herself how useful the data used in the paper is. Considering the spatial resolution given in the said website the usefulness is doubtful. As an example of interest for the reviewer: the country of Sweden is presented as one "basin" !

Thank you for raising this issue. Perhaps the simple website address is not sufficient as a reference for this data, and that will be amended. Specific information on the creation of the database can be accessed through hitting on the 'i' in the upper right corner of the cited website, which leads to this page describing the models:

http://floods.wri.org/#/

This page cites the sources as:

Winsemius, H. C., et al. "A framework for global river flood risk assessments." Hydrology and Earth System Sciences 17.5 (2013): 1871-1892.

and

Ward, P. J., Jongman, B., Weiland, F. S., Bouwman, A., van Beek, R., Bierkens, M. F., ... & Winsemius, H. C. (2013). Assessing flood risk at the global scale: model setup, results, and sensitivity. Environmental research letters, 8(4), 044019.

As the first of these papers is in fact published by HESS, we have deemed this data to be trustworthy.

Further papers published with this data include:

Winsemius, Hessel C., et al. "Global drivers of future river flood risk." Nature Climate Change 6.4 (2016): 381-385.

Jongman, B., Winsemius, H. C., Aerts, J. C., de Perez, E. C., van Aalst, M. K., Kron, W., & Ward, P. J. (2015). Declining vulnerability to river floods and the global benefits of adaptation. Proceedings of the National Academy of Sciences, 112(18), E2271-E2280.

Muis, S., Güneralp, B., Jongman, B., Aerts, J. C., & Ward, P. J. (2015). Flood risk and adaptation strategies under climate change and urban expansion: A probabilistic analysis using global data. Science of the Total Environment, 538, 445-457.

Yes, it is a concern that Sweden is represented as a single basin in this database! We have chosen to work with the city information instead of basins, so fortunately this is basin does not appear in our analysis.

(p.4, lines 29-34) Another weakness in the method is that the actual, real flood protection level is not included. Instead flood protection level is based on the assumption of proportionality with national income level.

This is a shortcoming of the method, as the real flood protection levels for each city are not known (by us) for each city. However, the assumptions made to overcome this are in line with current practice, as outlined in lines 33-37 of p4 with references to contemporary literature:

'…we assign an assumed flood protection level to each city based on the country's World Bank income level (as in the World Resource Institute's Aqueduct Global Flood Risk Country Rankings, website 6) due to a lack of information on each city's actual protection level. This method follows recommendations based on the rational that higher standards of protection against flooding may be expected in higher income countries (Jongman et al., 2012; Nicholls et al., 2008), and findings by Doocy et al. (2013) that flood impacts are significantly associated with classification of income level by the World Bank. '

This method is necessary as standardised, current information on the nearly 100 cities in the data set is not readily available. This shortcoming is acknowledged in the literature, as Winsemius et al. (2016) state that 'currently installed flood protection is an important missing link in the assessment of global flood risk'. If there is a better practise that we have missed, we would be very happy to learn of it.

(p.4, line 18-22, 38) On the one hand: changes due to socioeconomic development are driven by population and economy. On the other hand: protection level, which is assumed proportional to national income level is kept constant over time. This is inconsistent.

True, this is inconsistent. However, as discussed above the current flood protection levels are estimates, and therefore it seemed reasonable to retain them since the changes in flood protection in the 20 years of the study would also be estimates, however these would be based on estimated income levels, increasing the uncertainty.

(p. 6, lines 16-19) The present situation is in the paper characterized based on conditions 2010 while effects of climate change is based on projections for 2030. This time interval, 20 years, is a bit short for meaningful comparisons.

Whilst this is a short timeline, it is restricted by the data that is available. Studies at a global scale have been traditionally limited due to lack of cohesive data sets, and therefore this data set is valuable for the fact that it spans a global set of cities and provides an opportunity for comparison. As the data is only provided for 2010 and 2030, there was no choice for us to use different dates. Whilst it may not provide a long-term outlook, at the very least an insight into the current and near-future conditions can be gained.

(p.8, 3 lines 5-15; Fig. 1a) While the colouring of the maps allows for different absolute impact on people and material damage for an individual city, the location of all cities are exactly the same on both people and damage maps. This latter fact seems to indicate that the relative difference in impact between cities remains almost identically the same for both people and damage.

The relative placement of the cities on the map is the main map characteristic providing insight into the features of the data. The map is created based on both population and material damage data, organizing the cities with respect to each other based on both of these factors. The same map is repeated in Fig. 1a, with the background colouring indicating levels of population damage (on top left) and material damage (on top right). Therefore, the relative distance between cities on the map indicates the differences in a *combination* of population and material damages (which can be discerned from the colouring). For example, Cincinnati (top right) incurs high material damages costs, and medium population affected, whereas Ulan Bator (mid left) has similar population affected to Cincinnati, but much lower material damages costs.  This interpretation is explained on p8 in lines 5-23.

(p.17, lines (23-38) The actual, on the ground flood management measures and other socioeconomic development are presented for Marrakech. However, in the model producing the source data for this paper only some of these factors were incorporated. See previous comment re page 4.

The analysis in this paper is based solely on the data provided. We are not aware of the extent to which the on-the-ground flood management measures are incorporated into the socioeconomic models which produced this data. The discussion on p17 is included as a point of interest to help characterise a couple of individual cities, providing possible reasons why they may fit into the map where they do, for the readers' interest.

**Technical corrections:**

(p.1) The title is"…..changes on river…"should be"…..changes in river…"

(p.7: 25) "…..Whist…" should be "…..Whilst …"

(Fig. 1) Fig. 1a) is difficult to read because of the many cities and the small font; however still ok.

Fig. 1b) is too blurry due to combination of background colours and text. More or less impossible printed; still difficult on the screen.

(Fig.2) Similar problems as with Fig. 1

(Fig.3) Similar problems as with Fig. 1 &2. Difficulties reading this figure exacerbated by all the information crammed into one page. The gradients indicated along the axes are impossible to interpret.

(p.13:1) "…..reduction on…" should be "…..reduction of …"

P.16:16) "…..loses…" should be "…..losses …"

Thank you for pointing out these technical corrections, they will be addressed in future submissions.

---

## Author Comment (AC4) · 27 Sep 2017

Dear Dr. Toth,

Thank you for the opportunity to submit a revised manuscript. We appreciate all the comments we have received and have made a great effort to incorporate each of them into our revision.

As suggested, the manuscript has been thoroughly reconsidered and amended to provide better clarity on both the method and data used in this analysis. In particular, more details have been added with respect to the method and data, as follows:

- The Methods section has been expanded to include more specific information on the SOMs method (with equations), as have the error measures and Davies-Bouldin index.
- Data sources have been appropriately cited, and a list of previous papers produced using the same data has been provided. The timeline of the study, interpretation of the placement of the cities on the maps, and the necessary assumption of flood protection level have been further discussed.

All points recommended by the reviewers and editor have been systematically addressed and responded to. Please see the 'response to reviewer' documents submitted concurrently for a detailed response to each query about the method and/or data. We hope the information added to the Data and Methods sections, as well as further detail in the Introduction and Conclusion, will remove any further confusion regarding the pertinence of the method and validity of the data in this study.

The vocabulary used in the manuscript has been checked and updated. The term 'spatiotemporal' has been removed, and other terms mentioned by reviewer 1 have also been revised. The references have been updated as suggested.

A few paragraphs have been added to the end of the introduction to explicitly describe what types of 'patterns' and 'clusters' are expected to be found in this study and why they are of interest. This is also reiterated at the beginning of the Discussion. The conclusion has been expanded to include more discussion of shortcomings of this analysis, and areas that could be improved in future studies.

We hope you find this revised manuscript much improved.

Kind regards,

Stephanie Clark

---

## Author Comment (AC5) · 27 Sep 2017

Reviewer 1:

We thank you for your time and effort in providing comments on this paper.

- Specific changes that have been made to the revised manuscript based on these comments are described in the dot points following each comment.

The paper explores the global dataset on fluvial flood risk prepared and discussed by Philip Ward and co-authors on city level. This is potentially a good idea; although the data has been explored in a series of papers by these authors there are still room for further studies. In its scope the current paper is rather close to Winsemius et al (2016); the current paper focuses on cities while the original paper focuses on river catchments.

As mentioned, Winsemius et al (2016) analyses the data set with respect to river basin and income level. Our analysis is based on city-level flood impacts. Whilst similar, these studies are not the same, and the insights revealed in this paper do not overlap with those provided in Winsemius et al (2016).

- This is already stated in the Introduction, paragraph 5.

There are however a number of important shortcommings that should be addressed before I potentially can recommend publication.

First of all I would like to see a discussion of what new knowledge the authors think they can gain by considering cities rather than river catchments. At least the findings of the current paper should be compared to Winsemius et al (2016).

The comparison of cities allows an understanding of conditions that can be expected on a city-level, and identification of cities facing similar circumstances. As climate change, development, and urban administrations transcend river basin boundaries, it is sensible to investigate impacts and determine potential mitigation strategies at the city level (as well of course as basin level, but that has been covered in Winsemius et al (2016)).

This is discussed in the conclusion of this paper, p19 lines 12-26, which is reproduced here:

'This study adds to the understanding of natural hazards in a global context, which is an important aspect of regional disaster risk management due to the dependency of local situations on global processes (UNISDR, 2015). The complex nonlinear socio-environmental relationships make it difficult to foresee local responses to global changes (UNISDR, 2015), and therefore this study focuses on risk communication (the process between risk perception and adaptation planning (Cardona et al., 2012)) to provide a visual analysis of the global patterns of evolving flood impacts, socioeconomic development and climate change, and the local city-level consequences of these changes.

Cities have major implications for climate change mitigation and adaptation (Revi et al., 2014). Unplanned development and urban migration are increasing vulnerabilities to natural hazards (UNEP, 2016) and land cover change and greenhouse gas emissions are intensifying urban hydrology. Understanding the relationship between flood impacts and social vulnerability is a necessary step for prioritizing flood mitigation and prevention strategies (Doocy et al., 2013). Whether the main driver of increased urban flood impacts is development or climate change, cities will benefit from development restrictions and planning standards for urban expansion, sustainable land development, management of population distribution and migration, and early warning systems and preparedness (Revi et al., 2014; UN-DESA, 2014; Doocy et al., 2013).'

- A discussion of the interest of flood impacts in cities already exists in the conclusion, and this point has also been added to the final paragraph of the introduction.

Next, the introduction is rather long and gives a thorough introduction to flooding. Unfortunately the authors does not distinguish between the different types of flooding that can occur, i.e. sea surges, groundwater induced flooding, pluvial flooding and, fluvial flooding.

It is stated on p4, lines 13-14, that this paper only considers fluvial flooding:

'This data is solely related to the influence of fluvial flooding on metropolitan areas, and does not include coastal or pluvial flooding.'

If this statement would be better placed in the introduction, it can be moved, however we are conscious that the introduction is already 'rather long'.

- A sentence has been added to the Introduction, in paragraph 4, to reiterate that only fluvial flooding is considered. The phrase 'river flooding' is used seven times in the abstract and introduction.

The paper is based on a specific data set that only considers fluvial flooding. Hence references to authors that specifically refer to pluvial flooding should be removed, e.g. Willems et al (2012).

We refer to Willems a few times, for example in the following statement (p2 lines 36-37):

'Increases in rainfall intensity at urban hydrology scales of up to 60% are anticipated by 2100 (Willems, 2012).'

Though Willems may be generally interested in pluvial flooding, any references made here concern Willems' statements on increases in rainfall, not flooding.

Perhaps differences in fluvial flood risk between cities can be explained by different exposures to other types of flood risk? If the authors do not wish to enter such discussions they should stick to considering only one type of flooding.

The comment 'if the authors do not wish to enter such discussions they should stick to considering only one type of flooding' is confusing for three reasons.

First, we have made it clear we are only considering one type of flooding - fluvial flooding (see previous comment).

Second, we have 'entered such discussions'. The exposures of various cities to other types of flood risk is discussed in the 'discussion' section of the paper (eg. p18 lines 22-28, as reproduced here). Unfortunately, the reviewer has stated (below) that they have not read this part of the paper.

'Though this study does not consider coastal flooding, it may be noted that due to their locations near river mouths, many of the cities in the lower left of the map that are projected to experience high increases in impacts from river flooding are also at risk of increased coastal flooding from intensified storms and sea level rise due to climate change. Mumbai, Guangzhou, Shanghai, Ho Chi Minh City, Kolkata, Bangkok, and Dhaka are 7 of the top 14 cities (out of 136) ranked by current population exposure to coastal flooding. These same cities also comprise the top 7 cities (in this order: Kolkata, Dhaka, Mumbai, Guangzhou, Ho Chi Minh City, Shanghai, Bangkok) ranked by future (2070) estimated population exposed to coastal flooding (UNEP, 2016; Nicholls et al., 2008).'

Third, the reviewer has implied in a previous comment that it is unfortunate different types of flooding are not distinguished: 'Unfortunately the authors does not distinguish between the different types of flooding that can occur, i.e. sea surges, groundwater induced flooding, pluvial flooding and, fluvial flooding.'

Therefore, it is unclear what the reviewer is expecting to result from this comment.

Next, the literature on whether flood risk is stationary or has an increasing trend is quite abundant. The findings differ, primarily as a function of the framing, i.e. if the models include corrections for changes in socio-economic development, vulnerability, etc. The author uses the terms 'impacts', 'risks' and 'material damage' more or less inter-changeably throughout the paper. When referring to IDSR and other recognized frameworks I would expect more stringent use of the terminology and more transparent explanations and assumptions.

- In paragraph 3 of the Data section, we have indicated that one variable of the data set will be referred to in this paper as 'damages'. It has been defined here as 'urban property damages costs (in US dollars)', and this term has only been used for this.
- Use of the term 'risk' (dictionary definition: *To expose (someone or something valued) to danger, harm, or loss. The possibility that something unpleasant or unwelcome will happen*.) has been checked within the manuscript and we believe it has been used adequately in each instance. If there is a specific occurrence we should reconsider, please list it explicitly.
- Use of the term 'impacts' (dictionary definition: *A marked effect or influence. Have a strong effect on someone or something*.) has been checked within the manuscript. This term has been used in the manuscript to describe the influence that flooding will have on the population and urban property. In some instances, it could perhaps be replaced with 'risk' but in these cases we believe the word 'impact' also fits. If there are specific occurances of use the reviewer would prefer changed, please list them as we are happy to attempt for the greatest clarity possible.

This includes a description of what the scenario for development is between 2010 and 2030 and the rationale for choosing the approximately 100 cities in Table 1, that are then later reduced to 80 cities (without mentioning which of the cities are excluded).

The development scenarios are discussed in the Data section (p4 lines 18-28) with references to the models that were used to produce the data and accompanying documentation. The specific scenario is not described explicitly, however enough detail is provided for an interested reader to find the specific details:

'Three separate scenarios of climate change and socioeconomic development (optimistic, business-as-usual, and pessimistic) are given in Aqueduct, and in this study we use data from the business-as-usual case for our future flood impact scenario.'

The rationale for choosing the cities is explicitly given on p4, lines 6-8:

'The selection of cities used here is based on a list provided by the Lincoln Institute of Land Policy's Atlas of Urban Expansion (Angel et al., 2010, website 1), spanning all continents except Antarctica, encompassing four economic levels and four population levels.'

The reason that some of these cities have been excluded is explicitly state on p5, lines 5-6:

'Cities with no flood impacts in both 2010 and 2030 were removed (22 cities), though cities with no flood impacts in 2010 but with flood impacts in 2030 have been kept in the study.'

The reviewer is wondering why there is no mention of which cities were excluded - it was decided that a list of cities that are NOT included in this study would not be of interest to the readers hence they have not been listed.

- The reference to Table 1 in the text has been moved until after the discussion as to why some cities were not included. Hopefully this will make it obvious that only the cities in Table 1 are included in this study.

However, my most important concern is that I cannot see what the authors are doing with the data. The closest to an explanation is that the authors state that the calculations are carried out in Matlab using a modified version of the package SOM. Modified in which way?

The entire Methods section is describing what we have done with the data. Clearly the use of the SOM is causing some confusion, so this will be elaborated on.

- The description of the SOM method has been expanded to include equations and more detailed explanations.

Why are the data transformed the way they are if the SOM approach is particularly good in dealing with non-linear relationships,

The data is log transformed, following Agarwal and Skupin's (2008) recommendation that highly skewed variable distributions may benefit from log transformation before use in the SOM. *[Agarwal, P. and A. Skupin (2008). Self-organising maps: Applications in geographic information science, John Wiley & Sons.]* This information and reference will be added to the methods section.

- This information and reference have been added to the end of the data section.

how do you treat taking logarithm of the value of zero,

The treatment of the logarithm of values between -1 and 1 is carefully described on p15, lines 12-13.

what are the definitions of QE, TE, DRR, and the Davies-Bouldin Index,

QE, TE, and DRR are defined on p7, lines 16-19. A reference is provided for the Davies-Bouldin Index with the information that it is used to determine the number of clusters (p7, line 31).

- These definitions have been expanded with further explanations and equations.

and how is a U-matrix visually verified? Without this information the reader will have to do a complete reanalysis to (perhaps) get the same results as the authors.

Further SOM references have been provided for readers who are unfamiliar with the method, though it has been summarised in the Methods section. In an effort to avoid repetition of previously published material, the SOM method has not been reproduced in detail. As the papers and books outlining the SOM method have already been cited over 40,000 times, it was not felt that this method needed further repetition. With any paper, it is hard to know how much background knowledge to expect the readers to possess, and therefore we provide direction to further reading for those unfamiliar with certain aspects of the background.

- A paragraph has been added further describing the U-matrix.

Since I expect a thorough revision of the method section before perhaps resubmitting the paper I have not read the results and discussions sections in detail.

We are disappointed to hear this. I expect the reviewer would have found some answers to their questions had they read the paper.

**More detailed comments:**

I disagree with the use of the term 'spatio-temporal'. It usually denotes something where there is explicit reference to a spatial dimension, in the current example perhaps the physical distance between the cities. I think the authors should find another term to describe the characteristics of their data.

The term 'spatiotemporal' is commonly used to refer to data in which there is a cross-sectional structure to the data as well as a temporal one. In this case these are: 1) the state of flooding at each timestep, 2) and the timesteps themselves. The spatial dimension does not always refer to geographic distance.

- The term spatiotemporal no longer appears in the manuscript.

The data on the figures cannot be read in the pdf-version of the paper.

- Figures 1b), 2b), 3a) and 3b) have been reproduced with larger text, and the background colours have been lessened in intensity to make the text appear more clearly. We have tried a variety of text sizes on Figures 1a) and 2a), and believe the current text size makes it more clear precisely where each city is located on the map than a larger text size would. A note has been added to the caption referring the reader to the online version which can be zoomed in on. Figure b) maps with larger text could also be referred to if there is an issue reading the a) maps. The following is a screen shot of the pdf version of 1a), first at actual size, and then zoomed in. We believe both versions are readable.

[Figure]

Amount of population affected

I cannot follow the discussion on the cluster. Perhaps it is just me not being able to see the same patterns as the authors.

- Hopefully the lack of clarity on 'clusters' has been resolved by the addition of information regarding the clustering with the SOM that has been added to the Method section, the Discussion and in other locations throughout the manuscript. Please see the 'response to reviewer 2' for more info on this.

The list of references should be improved. Just from browsing it I can see dubious referencing to e.g. IPCC (2014) (Use author names) and Willems (2012) (several authors), and Kohonen (2001) (incomplete reference). There are more errors than the ones I have pointed out.

- The references have been checked and updated where necessary. The following, and others, have been updated as suggested:
  - Pachauri, RK, Allen, MR, Barros, VR, Broome, J, Cramer, W, Christ, R, … & Dubash, NK. (2014). Climate Change 2014: Synthesis Report. Contribution of Working Groups I, II and III to the Fifth Assessment Report of the Intergovernmental Panel on Climate Change (p. 151). IPCC.
  - Willems, P, Olsson, J, Arnbjerg-Nielsen, K, Beecham, S, Pathirana, A, Gregersen, IB, & Madsen, H. (Eds.). (2012). Impacts of climate change on rainfall extremes and urban drainage systems. IWA publishing.
  - Kohonen, T. (2001). Self-organizing maps, Volume 30 of Series in Information Sciences.

---

## Author Comment (AC6) · 27 Sep 2017

Reviewer 2:

Thank you very much for your time in reviewing this paper, Dr Larsson. We appreciate all comments and will address each one as thoroughly as possible to produce an improved paper. It appears your main concern rests with the data source used in the paper. We understand that more information is needed in the referencing of this data set, and hope that the information provided below will allay these concerns.

- Specific changes that have been made to the revised manuscript based on these comments are described in the dot points following each comment.

**General comments:**

The topic of this paper is clearly both interesting and of great importance. The paper is well written and has a clear structure.

However, it seems to me that the paper does not really fulfil the promises implicit in the title. Which are the patterns in flood impacts on cities that are revealed in the paper?

The 'patterns' are the key characteristics extracted for each cluster of cities sharing similar baseline and projected flood conditions. Each city in each cluster is matched to the 'pattern' represented by the cluster centroid. As described on p6 lines 1-3, the values of the prototype vectors (map nodes) come to represent the prevalent patterns in the data after they have self-organized amongst the data items. For example, cities in cluster 10 in the upper right of Fig.3 share the 'pattern' of low baseline flooding impacts, with small increases in population impacts projected primarily due to climate change, and increases in material damages projected due to both climate change and development. For Fig3, the text on p13 and 15, and Fig 4 on p14 interpret these patterns in detail. The Method section (p6) may benefit from expansion, as it appears more detail is needed on the process of pattern extraction by the SOM.

- A further description of the 'patterns' expected to be revealed with this analysis has been added to the final paragraph of the introduction. In the Methods section, the terms 'pattern' and 'cluster' have been defined in terms of the SOM method (which has been expanded upon), as well as the use of the terms with respect to this study. A paragraph has been added to the beginning of the Discussion section to again describe how the terms 'pattern' and 'cluster' relate to the results of this analysis. Hopefully this will clarify any confusion regarding references to 'patterns'.

There are some problems with the method used, and there is a problem with the general approach. The latter problem is related to the presumption that it is possible to draw conclusions about individual cities based on a global model. It is also doubtful whether any meaningful patterns can emerge from such a rather superficial study. The method issues are covered below.

It is not within the scope of our study to assess whether the model used to produce the data is comprehensive, but rather to provide a visualisation of the available data. The data set as published is immense and difficult to comprehend without a reduction and visualisation, which we are providing. Any conclusions about individual cities are based solely on the data items presented, and comparisons are based on similarities between data items, be them cities or merely numeric vectors.

- General issues with the method will hopefully be cleared up as we have added a significant amount of explanation and equations to the Methods section to better describe the SOM for readers unfamiliar with this method.

**Specific comments:**

(p.3, lines41-44) The objective is given only in rather general terms. It would have helped the reader to get some more precise information about which types of "global patterns and relationships" that is meant to emerge from the study.

This sentence had previously been more long-winded, but was reduced to this phrase for brevity with the knowledge that the 'patterns and relationships' would be explained in detail throughout the paper. This sentence can certainly be expanded to detail the expected patterns and relationships.

- The final paragraph of the introduction (now the second-last paragraph) has been expanded to better describe the 'patterns and relationships' that are expected to emerge from the study, with specific examples of patterns and relationships described.

(p.4, lines 11-20) The main problem with the method is the data source used in the paper. This source consists of output from a global tool produced by the World Resources Institute. In the paper this model is described as made up of "global hydrologic and hydraulic models" and more. It raises some concern about the accuracy of such models when they are "global". Unfortunately, the only reference in the paper leads to a web site, which in turn refers to the name of a model but with no proper literature reference. So, it is quite difficult for the reader to judge for herself how useful the data used in the paper is. Considering the spatial resolution given in the said website the usefulness is doubtful. As an example of interest for the reviewer: the country of Sweden is presented as one "basin" !

Thank you for raising this issue. Perhaps the simple website address is not sufficient as a reference for this data, and that will be amended. Specific information on the creation of the database can be accessed through hitting on the 'i' in the upper right corner of the cited website, which leads to this page describing the models:

http://floods.wri.org/#/

This page cites the sources as:

Winsemius, H. C., et al. "A framework for global river flood risk assessments." Hydrology and Earth System Sciences 17.5 (2013): 1871-1892.

and

Ward, P. J., Jongman, B., Weiland, F. S., Bouwman, A., van Beek, R., Bierkens, M. F., ... & Winsemius, H. C. (2013). Assessing flood risk at the global scale: model setup, results, and sensitivity. Environmental research letters, 8(4), 044019.

As the first of these papers is in fact published by HESS, we have deemed this data to be trustworthy.

Further papers published with this data include:

Winsemius, Hessel C., et al. "Global drivers of future river flood risk." Nature Climate Change 6.4 (2016): 381-385.

Jongman, B., Winsemius, H. C., Aerts, J. C., de Perez, E. C., van Aalst, M. K., Kron, W., & Ward, P. J. (2015). Declining vulnerability to river floods and the global benefits of adaptation. Proceedings of the National Academy of Sciences, 112(18), E2271-E2280.

Muis, S., Güneralp, B., Jongman, B., Aerts, J. C., & Ward, P. J. (2015). Flood risk and adaptation strategies under climate change and urban expansion: A probabilistic analysis using global data. Science of the Total Environment, 538, 445-457.

Yes, it is a concern that Sweden is represented as a single basin in this database! We have chosen to work with the city information instead of basins, so fortunately this is basin does not appear in our analysis.

- This information and these references have been added to the data section.

(p.4, lines 29-34) Another weakness in the method is that the actual, real flood protection level is not included. Instead flood protection level is based on the assumption of proportionality with national income level.

This is a shortcoming of the method, as the real flood protection levels for each city are not known (by us) for each city. However, the assumptions made to overcome this are in line with current practice, as outlined in lines 33-37 of p4 with references to contemporary literature:

'…we assign an assumed flood protection level to each city based on the country's World Bank income level (as in the World Resource Institute's Aqueduct Global Flood Risk Country Rankings, website 6) due to a lack of information on each city's actual protection level. This method follows recommendations based on the rational that higher standards of protection against flooding may be expected in higher income countries (Jongman et al., 2012; Nicholls et al., 2008), and findings by Doocy et al. (2013) that flood impacts are significantly associated with classification of income level by the World Bank. '

This method is necessary as standardised, current information on the nearly 100 cities in the data set is not readily available. This shortcoming is acknowledged in the literature, as Winsemius et al. (2016) state that 'currently installed flood protection is an important missing link in the assessment of global flood risk'. If there is a better practise that we have missed, we would be very happy to learn of it.

- A paragraph discussing this shortcoming has been added to the conclusion.

(p.4, line 18-22, 38) On the one hand: changes due to socioeconomic development are driven by population and economy. On the other hand: protection level, which is assumed proportional to national income level is kept constant over time. This is inconsistent.

True, this seems inconsistent. However, 'changes due to socioeconomic development' by nature consist of changes in population and economy, which are thoroughly measurable for current conditions and reliably estimated for future scenarios based on comprehensive models. On the other hand, the current flood protection levels are estimates and the changes in flood protection in the 20 years of the study would also be estimates, based on estimated projections of income levels (which are not readily available), increasing the level of uncertainty in a variable value that is already an assumption. Therefore, we have deemed it reasonable to retain a constant flood protection level for each city over the 20 years.

(p. 6, lines 16-19) The present situation is in the paper characterized based on conditions 2010 while effects of climate change is based on projections for 2030. This time interval, 20 years, is a bit short for meaningful comparisons.

Whilst this is a short timeline, it is restricted by the data that is available. Studies at a global scale have been traditionally limited due to lack of cohesive data sets, and therefore this data set is

valuable for the fact that it spans a global set of cities and provides an opportunity for comparison. As the data is only provided for 2010 and 2030, there was no choice for us to use different dates. Whilst it may not provide a long-term outlook, at the very least an insight into the current and near-future conditions can be gained.

- This is a good point you have made, and we have added a discussion of it in the conclusion.

(p.8, 3 lines 5-15; Fig. 1a) While the colouring of the maps allows for different absolute impact on people and material damage for an individual city, the location of all cities are exactly the same on both people and damage maps. This latter fact seems to indicate that the relative difference in impact between cities remains almost identically the same for both people and damage.

The relative placement of the cities on the map is the main map characteristic providing insight into the features of the data. The map is created based on both population and material damage data, organizing the cities with respect to each other based on both of these factors. The same map is repeated in Fig. 1a, with the background colouring indicating levels of population damage (on top left) and material damage (on top right). Therefore, the relative distance between cities on the map indicates the differences in a *combination* of population and material damages (which can be discerned from the colouring). For example, Cincinnati (top right) incurs high material damages costs, and medium population affected, whereas Ulan Bator (mid left) has similar population affected to Cincinnati, but much lower material damages costs. This interpretation is explained on p8 in lines 5-23.

- This explanation has been added here.

(p.17, lines (23-38) The actual, on the ground flood management measures and other socioeconomic development are presented for Marrakech. However, in the model producing the source data for this paper only some of these factors were incorporated. See previous comment re page 4.

The analysis in this paper is based solely on the data provided. We are not aware of the extent to which the on-the-ground flood management measures are incorporated into the socioeconomic models which produced this data. The discussion on p17 is included as a point of interest to help characterise a couple of individual cities, providing possible reasons why they may fit into the map where they do, for the readers' interest.

- A paragraph has been added after paragraph 4 of the discussion to explain this, and references have been added to the data to reinforce this point.

**Technical corrections:**

(p.1) The title is".....changes on river..."should be".....changes in river..."

- Good point! This has been changed, thank you.

(p.7: 25) ".....Whist..." should be ".....Whilst ..."

- Thank you, this has been updated.

(Fig. 1) Fig. 1a) is difficult to read because of the many cities and the small font; however still ok. Fig. 1b) is too blurry due to combination of background colours and text. More or less impossible printed; still difficult on the screen.

- Figures 1b), 2b), 3a) and 3b) have been reproduced with larger text, and the background colours have been lessened in intensity to make the text appear more clearly. We have tried

a variety of text sizes on Figures 1a) and 2a), and believe the current text size makes it more clear precisely where each city is located on the map than a larger text size would. A note has been added to the caption referring the reader to the online version which can be zoomed in on. Figure b) maps with larger text could also be referred to if there is an issue reading the a) maps. The following is a screen shot of the pdf version of 1a), we believe it is readable.

[Figure]

(Fig.2) Similar problems as with Fig. 1

- Figures 1b), 2b), 3a) and 3b) have been reproduced with larger text, and the background colours have been lessened in intensity to make the text appear more clearly.

(Fig.3) Similar problems as with Fig. 1 &2. Difficulties reading this figure exacerbated by all the information crammed into one page. The gradients indicated along the axes are impossible to interpret.

- Figures 1b), 2b), 3a) and 3b) have been reproduced with larger text, and the background colours have been lessened in intensity to make the text appear more clearly. A screen shot of the pdf of Figure 3a) is shown below. Though the gradients aren't too large, a general idea of their magnitude in the different regions of the map can be discerned, which was the intention. The figure has been reformatted with the legend at the bottom, which has

allowed the figure room to expand sideways on the page, making it larger. Again, a note has been added that the reader can refer to the online version to zoom in on the figure.

[Figure]

(p.13:1) "…..reduction on…" should be "…..reduction of …"

- Thank you, this has been updated.

P.16:16) "…..loses…" should be "…..losses …"

- Thank you, this has been updated.

---

## Author Comment (AC8) · 27 Sep 2017

[revised manuscript text omitted]
       |                        |              | Zugdidi            | Georgia                       |  |
|--------------------|---------------|------------------------|--------------|--------------------|-------------------------------|--|
| Anqing             | China         | South Asia             |              |                    |                               |  |
| Ansan              | Rep. of Korea | Dhaka                  | Bangladesh   | North Africa       |                               |  |
| Beijing            | China         | Hyderabad              | India        | Alexandria         | Egypt                         |  |
| Changzhi           | China         | Jalna                  | India        | Algiers            | Algeria                       |  |
| Chinju             | Rep. of Korea | Kanpur                 | India        | Aswan              | Egypt                         |  |
| Fukuoka            | Japan         | Kolkata                | India        | Cairo              | Egypt                         |  |
| Guangzhou          | China         | Mumbai                 | India        | Casablanca         | Morocco                       |  |
| Leshan             | China         | Puna                   | India        | Marrakech          | Morocco                       |  |
| Pusan              | Rep. of Korea | Rajshahi               | Bangladesh   | Port Sudan         | Sudan                         |  |
| Seoul              | Rep. of Korea | Vijayawada             | India        | Tebessa            | Algeria                       |  |
| Shanghai           | China         |                        |              |                    |                               |  |
| Sydney             | Australia     | Western & Central Asia |              | Sub-Saharan Africa |                               |  |
| Tokyo              | Japan         | Ahvaz                  | Iran         | Accra              | Ghana                         |  |
| Ulan Bator         | Mongolia      | Astrakhan              | Russian Fed. | Bamako             | Mali                          |  |
| Yiyang             | China         | Baku                   | Azerbaijan   | Harare             | Zimbabwe                      |  |
| Yulin              | China         | Gorgan                 | Iran         | Ibadan             | Nigeria                       |  |
| Zhengzhou          | China         | Istanbul               | Turkey       | Johannesburg       | South Africa                  |  |
|                    |               | Kuwait City            | Kuwait       | Kampala            | Uganda                        |  |
| Southeast Asia     |               | Malatya                | Turkey       | Kigali             | Rwanda                        |  |
| Bandung            | Indonesia     | Moscow                 | Russian Fed. | Ouagadougou        | Burkina Faso                  |  |
| Bangkok            | Thailand      | Oktyabrsky             | Russian Fed. | 0 0                |                               |  |
| Ho Chi Minh City   | Vietnam       | Sanaa                  | Yemen        |                    |                               |  |
| Kuala Lumpur       | Malaysia      | Shimkent               | Kazakhstan   |                    |                               |  |
| Manila             | Philippines   | Teheran                | Iran         | Latin America & t  | Latin America & the Caribbean |  |
| Palembang          | Indonesia     | Tel Aviv               | Israel       | Buenos Aires       | Argentina                     |  |
| Songkhla           | Thailand      | Vorovan                | Armonia      | C                  | Veneruele                     |  |

[revised manuscript text omitted]

**Amount of population affected**

Shanghai Kolkata Cairo Seoul Paris Los Angeles Philadelphia Bangkok Buenos Aires Tokyo Chicago Cincinnati Mexico CityBuenos Aires Tokyo Phil Mumbai Dhaka Sao Post Manila Sao Post Palembang Tijuana Guangzhou Moscow Hyderabad Puna Kanpur Moscow Yulin Astrakhan Yulin Wa Houston Ahvaz London Minneapolis Tijuana Pittsburg Modesto Springfield

Ho Chi Minb City Bandung Kuala Lumpur Rajshahi Anqing Warsaw Tacoma Vijayawada Tel Aviv Yerevan Beijing Yiyang Lean Kigali

Chinju Algiers Shimkent Castellon Ulan Bator Port Sudan Oktyabrsky

Marrakech Gorgan Changzhi Leipzig Fukuoka Aswan Malatya Jaina Montevideo Alexandria Caracas Ouagadougou Harare Sheffield

Ibadan Pusan Guadalajara Sanaa Istanbul Baku Zugdidi Sydney Tebessa Thessaloniki Valledupar Ribeirao Preto Johannesburg Ansan St. Catharines Teheran Songkhla Kampala

Urban damages costs

Shanghai Kolkata Cairo Seoul Paris Los Angeles Bangkok Buenos Aires Tokyo Chromoti ngkok Phile Mexico CityBuenos Aires Tokyo Chicago Mumbal Anyaz Houston Dhaka Manila Sao Paulo London Anvaz Wien Manila Palembang Guangzhou Hyderabad Puna Kanpur Moscow Springfield Hyderabad Santiago Yulin Astrakhan Warsaw Minneapolis Tijuana PittsburgModesto Ho Chi Ming City Bamako Bandung Kuala Lumpur Warsaw Rajshahi Anqing Tac Vijavawada Tac Tacoma Beijing Yiyang Zhengzhou Vijayawada Yerevan Tel Aviv Chinju Algierscra Shimkent Kigali Castellon Ulan Bator Port Sudan Oktyabrsky Marrakech Gorgan Changzhi Leipzig Aswan Fukuoka Jaina Malatya Montevideo Alexandria Caracas Ouagadougou Harare Sheffield Ibadan Pusan Guadalaiara Sanaa Istanbul Baku Zugdidi Sydney Tebessa Thessaloniki Valledupar Ribeirao Preto Johannesburg Ansan Teheran St. Catharines Kampala

**High impact**

**GDP affected**

Low impact

Shanghai Kolkata Cairo Seoul Paris Los Angeles Bangkok Buenos Aires Tokyo Mexico City Buenos Aires Tokyo Mumbai Anyaz Chicago Houston Ahvaz Dhaka Manila Wien Sao Paulo Manila Palembang Guangzhou Hyderabad Casablanca Santiago Yulin London Minneapolis Tijuana PittsburgModesto Springfield Astrakhan

Ho Chi Minb City Bamako Bandung Kuala Lumpur Rajshahi Angin Warsaw Anging Tacoma Vijayawada Tel Aviv Vijayawada Beijing Yiyang Jequie Kigali Le Mans Chinju Algiers Cra Shimkent Castellor Castellon

Oktyabrsky Budapest Madrid Ulan Bator Port Sudan Marrakech Gorgan Changzhi Leipzig Aswan Fukuoka Malatya Montevideo Jalna

Alexandria Caracas Ouagadougou Harare Sheffield Sanaa Istanbul Ibadan Pusan Guadalajara Zugdidi Sydney Baku Tebessa Thessaloniki Valledupar Ribeirao Preto Johannesburg Ansan

Teheran St. Catharines Songkhla Kampala

**Percent of city's population affected**

Songkhla

Shanghai Kolkata Cairo Seoul Paris Los Angeles Bangkok Philadelphia Mexico City Buenos Aires Tokyo Chicago Chiconnati Houston Dhaka Palembang Guangzhou Hyderabad, Puna Kanpur Moscow Casablanca Santiago Yulin - Kuala Lump Mumbai Houston Ahvaz Wien ndon Minneapolis Tijuana Pittsburg Modesto Springfield Astrakhan

Ho Chi Minb City Bamako Bandung Kuala Lumpur Warsaw Rajshahi Ta Vijavawada Anqing Te Tacoma Vijayawada Beijing Yiyang Le Mans Vijayawada Tel Aviv Kigali Yerevan Shimkent City

Chinju Algiers Castellon Oktyabrsky Budapest Madrid Ulan Bator Port Sudan Leipzig

Marrakech Gorgan Changzhi Fukuoka Aswan Montevideo Jalna Malatya Alexandria Caracas Ouagadougou Harare Sheffield Ibadan Pusan Guadalajara Sanaa Istanbul

Baku Zugdidi Sydney Ribeirao Preto Tebessa Thessaloniki Valledupar Johannesburg Ansan

Teheran St. Catharines Songkhla Kampala

---

## Referee Report (RR1)

Rolf Larsson
Dept of Water Resources Engineering
Lund University, Sweden
8 November  2017

**Review of revised paper. Original review further below.**
§§§§§§§§§§§§§§§§§§§§§§§§§§§§§

Journal: HESS
Title: Patterns and comparisons of human-induced changes on river flood impacts in cities
Author(s): Stephanie Clark et al.
MS No.: hess-2017-162
MS Type: Research article

This is a review of a revised version of the paper. In the original review, some concerns were raised about the method(s) used. The revision has rectified some of the problems mentioned in the original review, while some problems remain. However, I suggest that the paper can be published.

Some comments, related to "specific comments" made in the original review:

- Objective. / OK now
- Data source. / New references have been added, which makes it easier for the reader to make up her mind about the accuracy of the so called global tool which has been used. HOWEVER, I am still doubtful about using the global hydrological model for all cities in the present study. In one of the references a validation is given for river basing larger than 150 000 km2. Several of the cities in the studies are located in much smaller river basins, e.g. London (Thames 16 000 km2) and Madrid (Manzanares 528 km2).
- Actual, real flood protection levels not used / REMAINS. This weakness in the model was identified already in one of the new references, Ward et al (2013).

Also: Some figures have been magnified and are now easier to read. HOWEVER, I still find Figures 1-3 non-reader-friendly.

§§§§§§§§§§§§§§§§§§§§§§§§§§§§§§§§§§§§§§§§§§§§§§§§§§§§§§

Rolf L

7 August 2017

§§§§§§§§§§§§§§§§§§§§§§§§§§§§§§

Journal: HESS
Title: Patterns and comparisons of human-induced changes on river flood impacts in cities
Author(s): Stephanie Clark et al.
MS No.: hess-2017-162
MS Type: Research article

This paper is accessible and currently open for Interactive Public Discussion at: https://www.hydrol-earth-syst-sci-discuss.net/hess-2017-162/#discussion. During this discussion, you are kindly asked to publish one or more Referee Comments; they can be anonymous or may be attributed and published under your name if you prefer. Please submit your comment at your earliest convenience but no later than 16 Aug 2017.

Interactive comments are published alongside the discussion paper and will remain permanently archived, publicly accessible and fully citable.

Generally a referee comment should be structured as follows: an initial paragraph or section evaluating the overall quality of the discussion paper ("general comments"), followed by a section addressing individual scientific questions/issues ("specific comments"), and by a compact listing of purely technical corrections at the very end ("technical corrections": typing errors, etc.).

§§§§§§§§§§§§§§§§§§§§§§§§§§§§§§§§§§§

**General comments**
The topic of this paper is clearly both interesting and of great importance. The paper is well written and has a clear structure. However, it seems to me that the paper does not really fulfil the promises implicit in the title. Which are the patterns in flood impacts on cities that are revealed in the paper?

There are some problems with the method used, and there is a problem with the general approach. The latter problem is related to the presumption that it is possible to draw conclusions about individual cities based on a global model. It is also doubtful whether any meaningful patterns can emerge from such a rather superficial study. The method issues are covered below.

**Specific comments**
(p.3, lines 41-44) The objective is given only in rather general terms. It would have helped the reader to get some more precise information about which types of "global patterns and relationships" that is meant to emerge from the study.

(p.4, lines 11-20) The main problem with the method is the data source used in the paper. This source consists of output from a global tool produced by the World Resources Institute. In the paper this model is described as made up of "global hydrologic and hydraulic models" and more. It raises some concern about the accuracy of such models when they are "global". Unfortunately, the only reference in the paper leads to a web site, which in turn refers to the name of a model but with no proper literature reference. So, it is quite difficult for the reader to judge for herself how useful the data used in the paper is. Considering the spatial resolution given in the said website the usefulness is doubtful. As an example of interest for the reviewer: the country of Sweden is presented as one "basin" !

(p.4, lines 29-34) Another weakness in the method is that the actual, real flood protection level is not included. Instead flood protection level is based on the assumption of proportionality with national income level.

(p.4, line 18-22, 38) On the one hand: changes due to socioeconomic development are driven by population and economy. On the other hand: protection level, which is assumed proportional to national income level is kept constant over time. This is inconsistent.

(p. 6, lines 16-19) The present situation is in the paper characterized based on conditions 2010 while effects of climate change is based on projections for 2030. This time interval, 20 years, is a bit short for meaningful comparisons.

(p.8, 3 lines 5-15; Fig. 1a) While the colouring of the maps allows for different absolute impact on people and material damage for an individual city, the location of all cities are exactly the same on both people and damage maps. This latter fact seems to indicate that the relative difference in impact between cities remains almost identically the same for both people and damage.

(p.17, lines (23-38) The actual, on the ground flood management measures and other socioeconomic development are presented for Marrakech. However, in the model producing the source data for this paper only some of these factors were incorporated. See previous comment re page 4.

**Technical corrections**
- (p.1) The title is "…..changes on river…" should be "…..changes in river…"
- (p.7: 25) "…..Whist…" should be "…..Whilst …"
- (Fig. 1) Fig. 1a) is difficult to read because of the many cities and the small font; however still ok. Fig. 1b) is too blurry due to combination of background colours and text. More or less impossible printed; still difficult on the screen.
- (Fig.2) Similar problems as with Fig. 1
- (Fig.3) Similar problems as with Fig. 1 &2. Difficulties reading this figure exacerbated by all the information crammed into one page. The gradients indicated along the axes are impossible to interpret.
- (p.13:1) "…..reduction on…" should be "…..reduction of …"
- P.16:16) "…..loses…" should be "…..losses …"

---

## Author Response (AR2)

**Editor Decision: Publish subject to minor revisions** (review by editor) (18 Dec 2017) by Elena Toth

Thank you very much for your comments. In addressing them, we have focused on highlighting the novel contributions of this study, as well as improving the readability of the manuscript.

*Responses to comments (in black) are provided in blue:*

1- even more emphasis should be put, and as soon as the city level is introduced as one of the main novelty of the work, on the strong limitations of an application at city scale (and with small catchment areas upstream , in some cases), partly due to the spatial scale of the applied models and partly due to the specific defence measures, that are much more important than at watershed scale, where, due to overall averaging, it may be more acceptable assuming them as proportional to the national income level.

A paragraph has been added to the introduction immediately after the description of the project to discuss limitations of the city-scale aspect. It begins 'The main analyses to date...'.

2- as said above, more stress, in abstract and Introduction, on the novelty of the application of the SOM approach (including the temporal extension to assess the expected changes in time) for analysing the Acqueduct dataset, which is indeed new.

Thank you for reminding us to focus on the importance of this paper as the first application of the SOM to this data. This is indeed the most important feature, and the abstract and introduction have been reworded to stress this. The SOM is now introduced in the first paragraph.

- as remarked by Ref#2, the introduction of the Acqueduct data set and the similarity and differences of the work with those by Ward et al and Winsemius et al, should be very clear since beginning, too.

The first paragraph of the introduction has been reworded to make it clear that this study is based upon the work of Winsemius and Ward and that the role of the current study is to analyse the Aqueduct city data through visualisation and clustering. This paragraph now explicitly states the motivation, data and novelty of the method at the very outset of the paper.

Following the above three points, the Introduction section is certainly to be deeply revised (and especially since more weight is to be given to the above points, I do agree with Ref#2 that the first two paragraphs of the current one may be summarised in a few lines).

The introduction has been extensively revised. The first paragraph now clearly explains the novelty of the study and the method used. The following paragraph summarise the climatic and developmental changes that are influencing the transformations in hydrology that are being explored. We feel it is important to retain some of this information to summarise the systems creating the data that is being analysed. Redundant information has been removed. (We have also thoroughly revised the results and discussion section to remove redundant information.)

The conclusions should stress more the potential and also the strong and weak points related to the application of the SOM networks, in order to better help the readers to understand how they be used also in other similar applications.

Sentences have been added to the conclusion to describe how the method used here could be adopted for other human-environmental                                              analysis                                              applications.

**Referee # 1: second set of comments:**

Thank you kindly for your comments on the paper. We have worked to revise the manuscript thoroughly to clarify the issues you have raised.

My basic question however remains: Is there sufficiently novelty in repeating the work by Ward et al by focussing on cities rather than catchments? If so, this should be clearly communicated. The authors mention that cities can benchmark against each other in the rebuttal letter. That is the closest to a justification of the work; however, the justification should appear in the paper.

A new paragraph has been added to the introduction specifically discussing the analysis of the data at the city scale, including limitations that exist to this analysis compared to one at basin scale. As suggested, the justification for a city scale analysis given in the previous rebuttal letter has been added to this paragraph.

Along the same line of thought also indicate the shortcomings of such an approach, that works in Australia, but not in other parts of the world, where cities are placed along rivers, whether it is the Elbe, the Meuse, or of course the very large rivers of Asia. There mitigation efforts in one city will impact downstream cities, making benchmarking quite difficult. If I were the editor such a short precise description of the justification and the novelty of the paper would be key to accepting the paper.

This has been included in the same paragraph added above.

A thing that I liked about the original manuscript was that it was clearly stated, that there was much similarity with the study by Winsemius et al (2016). Now you have to read the manuscript in detail to understand how closely related they are.

In the original manuscript, the study by Winsemius et al (2016) was referred to on page 3 line 25. In the previous version of the revised manuscript, the study by Winsemius et al (2016) was referred to on page 3 line 26 (in an unchanged sentence with respect to the original manuscript), as well as in four other locations in the manuscript. Discussion concerning the study by Winsemius et al (2016) had only been increased and nothing had been removed. In this revision, we have now also added a reference to Winsemius et al (2016) into the first paragraph of the introduction.

The paper is quite long and could be abbreviated without loss of information quite a few places. An example is the first two paragraphs in the Introduction that are redundant wrt information. Also discussion and conclusion are long and contain redundant information.

The first two paragraphs have been completely revised. The first now clearly indicates the aim of the project in analysing the Aqueduct data, why this is novel, and how it will be done. Following this, a single paragraph provides an overview of the hydrologic changes the data set deals with. It also defines what we are referring to as 'development' and how this influences hydrology. We feel this is necessary to make the paper accessible to a wide audience.

The discussion and conclusion have also been revised to remove redundant information and stress the importance of the information that is retained.

The structure should be further improved. In particular the definition of the measures (P8 L23-36) should be placed next to the description of the method, before the construction of the subsets are introduced, i.e. around P7 L 10.

As suggested, the description of the error measures has been moved immediately after the description of the method.

Discussion is very long, not very precise, and some of the text duplicates findings already found in the results section. The two best sections that should be in the discussion section are currently placed in the Conclusions section (P22 L21-32).

The discussion has been thoroughly revised, and information that was similar to that in the introduction has been removed. As suggested, P22 L21-32 have been moved from the conclusion to the discussion.

P10 L12: How does the non-linearity show up

This has been spelled out in more detail. The original sentence:

[revised manuscript text omitted]

Referee 1: Is there sufficiently novelty in repeating the work by Ward et al by focussing on cities rather than catchments? If so, this should be clearly communicated. The authors mention that cities can benchmark against each other in the rebuttal letter. That is the closest to a justification of the work; however, the justification should appear in the paper.

Along the same line of thought also indicate the shortcommings of such an approach, that works in Australia, but not in other parts of the world, where cities are placed along rivers, whether it is the Elbe, the Meuse, or of course the very large rivers of Asia. There mitigation efforts in one city will impact downstream cities, making benchmarking quite difficult. If I were the editor such a short precise description of the justification and the novelty of the paper would be key to accepting the paper.

[revised manuscript text omitted]